# WIDE-MINIMA DENSITY HYPOTHESIS AND THE EXPLORE-EXPLOIT LEARNING RATE SCHEDULE

## ABSTRACT

Several papers argue that wide minima generalize better than narrow minima. In this paper, through detailed experiments that not only corroborate the generalization properties of wide minima, we also provide empirical evidence for a new hypothesis that the density of wide minima is likely lower than the density of narrow minima. Further, motivated by this hypothesis, we design a novel explore-exploit learning rate schedule. On a variety of image and natural language datasets, compared to their original hand-tuned learning rate baselines, we show that our explore-exploit schedule can result in either up to 0.84% higher absolute accuracy using the original training budget or up to 57% reduced training time while achieving the original reported accuracy. For example, we achieve state-of-the-art (SOTA) accuracy for IWSLT'14 (DE-EN) and WMT'14 (DE-EN) datasets by just modifying the learning rate schedule of a high performing model.

## 1    INTRODUCTION

One of the fascinating properties of deep neural networks (DNNs) is their ability to generalize well, i.e., deliver high accuracy on the unseen test dataset. It is well-known that the learning rate (LR) schedules play an important role in the generalization performance (Keskar et al., 2016; Wu et al., 2018; Goyal et al., 2017). In this paper, we study the question, *what are the key properties of a learning rate schedule that help DNNs generalize well during training?*

We start with a series of experiments training Resnet18 on Cifar-10 over 200 epochs. We vary the number of epochs trained at a high LR of $0.1$, called the *explore* epochs, from 0 to 100 and divide up the remaining epochs equally for training with LRs of $0.01$ and $0.001$. Note that the training loss typically stagnates around 50 epochs with $0.1$ LR. Despite that, we find that as the number of explore epochs increase to 100, the average test accuracy also increases. We also find that the minima found in higher test accuracy runs are wider than the minima from lower test accuracy runs, corroborating past work on wide-minima and generalization (Keskar et al., 2016; Hochreiter & Schmidhuber, 1997; Jastrzebski et al., 2017; Wang et al., 2018). Moreover, what was particularly surprising was that, even when using fewer explore epochs, a few runs out of many trials still resulted in high test accuracies!

Thus, we not only find that an initial exploration phase with a high learning rate is essential to the good generalization of DNNs, but that *this exploration phase needs to be run for sufficient time, even if the training loss stagnates much earlier. Further, we find that, even when the exploration phase is not given sufficient time, a few runs still see high test accuracy values.*

To explain these observations, we hypothesize that, *in the DNN loss landscape, the density of narrow minima is significantly higher than that of wide minima.* A large learning rate can escape narrow minima easily (as the optimizer can jump out of them with large steps). However, once it reaches a wide minima, it is likely to get stuck in it (if the "width" of the wide minima is large compared to the step size). With fewer explore epochs, a large learning rate might still get lucky occasionally in finding a wide minima but invariably finds only a narrower minima due to their higher density. As the explore duration increase, the probability of eventually landing in a wide minima also increase. Thus, *a minimum duration of explore* is necessary to land in a wide minimum with high probability.

Heuristic-based LR decay schemes such as cosine decay (Loshchilov & Hutter, 2016) *implicitly* maintain a higher LR for longer than schemes like linear decay. Thus, the hypothesis also explains

cosine decay's better generalization compared to linear decay. Moreover, the hypothesis enables a principled learning rate schedule design that *explicitly* accounts for the requisite explore duration.

Motivated by the hypothesis, we design a novel *Explore-Exploit* learning rate schedule, where the initial *explore* phase optimizes at a high learning rate in order to arrive in the vicinity of a wide minimum. This is followed by an *exploit* phase which descends to the bottom of this wide minimum. We give *explore* phase enough time so that the probability of landing in a wide minima is high. For the *exploit* phase, we experimented with multiple schemes, and found a simple, parameter-less, linear decay to zero to be effective. *Thus, our proposed learning rate schedule optimizes at a constant high learning rate for some minimum time, followed by a linear decay to zero. We call this learning rate schedule the Knee schedule.*

We extensively evaluate the *Knee schedule* across a wide range of models and datasets, ranging from NLP (BERT pre-training, Transformer on WMT'14(EN-DE) and IWSLT'14(DE-EN)) to CNNs (ImageNet on ResNet-50, Cifar-10 on ResNet18), and spanning multiple optimizers: SGD Momentum, Adam, RAdam, and LAMB. In all cases, *Knee schedule* improves the test accuracy of state-of-the-art hand-tuned learning rate schedules, when trained using the original training budget. The explore duration is a hyper-parameter in *Knee schedule* but even if we set the explore duration to a fixed 50% fraction of total training budget, we find that it still outperforms prior schemes.

We also experimented with reducing the training budget, and found that *Knee schedule* can achieve the same accuracy as the baseline under significantly reduced training budgets. For the BERT$_{LARGE}$ pretraining, WMT'14(EN-DE) and ImageNet experiments, we are able to train in 33%, 57% and 44% less training budget, respectively, for the same test accuracy. This corresponds to significant savings in GPU compute, e.g. *savings of over 1000 V100 GPU-hours* for BERT$_{LARGE}$ pretraining.

The main contributions of our work are:

1. A hypothesis of lower density of wide minima in the DNN loss landscape, backed by extensive experiments, that explains why a high learning rate needs to be *maintained for sufficient duration* to achieve good generalization.
2. The hypothesis also *explains* the good performance of heuristic-based schemes such as cosine decay, and promotes a *principled design* of learning rate decay schemes.
3. Motivated by the hypothesis, we design an *Explore-Exploit* learning rate schedule called *Knee schedule* that outperforms prior heuristic-based learning rate schedules, including achieving state-of-the-art results in IWSLT'14 (DE-EN) and WMT'14 (DE-EN) datasets.

## 2    WIDE-MINIMA DENSITY HYPOTHESIS

Many popular learning rate (LR) schedules, such as the step decay schedules for image datasets, start the training with high LR, and then reduce the LR periodically. For example, consider the case of Cifar-10 on Resnet-18, trained using a typical step LR schedule of $0.1, 0.01, 0.001$ for 100, 50, 50 epochs each. In many such schedules, even though training loss stagnates after several epochs of high LR, one still needs to continue training at high LR in order to get good generalization.

For example, Figure 1 shows the training loss for Cifar-10 on Resnet-18, trained with a fixed LR of 0.1 (orange curve), compared to a model trained via a step schedule with LR reduced at epoch 50 (blue curve). As can be seen from the figure, the training loss stagnates after $\approx 50$ epochs for the orange curve, and locally it makes sense to reduce the learning rate to decrease the loss. However, as shown in Table 1, generalization is directly correlated with duration of training at high LR, with the highest test accuracy achieved when the high LR is used for 100 epochs, well past the point where training loss stagnates.

To understand the above phenomena, we perform another experiment. We train Cifar-10 on Resnet-18 for 200 epochs, using a high LR of $0.1$ for only 30 epochs and then use LR of $0.01$ and $0.001$ for 85 epochs each. We repeat this training 50 times with different random weight initializations. On an average, as expected, this training yields a low test accuracy of $94.81$. However, *in 1 of the 50 runs, we find that the test accuracy reaches $95.24$, even higher than the average accuracy of $95.1$ obtained while training at high LR for 100 epochs!*

**Hypothesis.** To explain the above observations, i.e., using a high learning rate for *short duration* results in low average test accuracy with rare occurrences of high test accuracy, while using the same

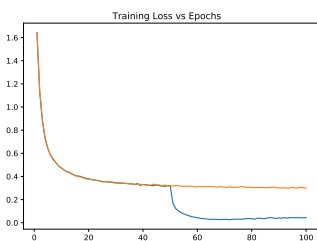

Figure 1: Training loss for Cifar-10 on Resnet-18. Orange plot uses a fixed LR of 0.1, while in blue plot, the LR is reduced from 0.1 to 0.01 at epoch 50.

Table 1: Cifar-10 on Resnet-18 trained for 200 epochs with Momentum. An LR of 0.1 is used for the explore epochs. Half the remaining epochs are trained at 0.01 and the other half at 0.001. Reported results are average over 4 runs.

| Epochs at 0.1 LR | Test Accuracy Avg. (Std. Dev) | Train Loss Avg. (Std. Dev.) |
|---|---|---|
| 0 | 94.34 (0.13) | 0.0017 (8e-5) |
| 30 | 94.81 (0.15) | 0.0017 (8e-5) |
| 40 | 94.91 (0.14) | 0.0018 (9e-5) |
| 60 | 95.01 (0.14) | 0.0018 (1e-4) |
| 80 | 95.05 (0.15) | 0.0019 (1e-4) |
| 100 | 95.10 (0.14) | 0.0021 (1e-4) |

high learning rate for *long duration* achieves high average test accuracy, we introduce a new hypothesis. We hypothesize that, *in the DNN loss landscape, the density of narrow minima is significantly higher than that of wide minima*.

An intuitive explanation of why high LRs are necessary to locate wide minima then follows: a large LR can escape narrow minima "valleys" easily (as the optimizer can jump out of them with large steps). However, once it reaches a wide minima "valley", it is likely to get stuck in it (if the "width" of the wide valley is large compared to the step size). For example, see Wu et al. (2018) for a result showing that large LRs are unstable at narrow minima and thus don't converge to them. Thus the optimizer, when running at a high LR, jumps from one narrow minimum region to another, until it lands in a wide minimum region where it then gets stuck. Now, the probability of an optimization step landing in a wide minima is a direct function of the proportion of wide minima compared to that of narrow minima. Thus, if our hypothesis is true, i.e., wide minima are much fewer than narrow minima, this probability is very low, and the optimizer needs to take a lot of steps to have a high probability of eventually landing in a wide minimum. This explains the observation in Table 1, where the average accuracy continues to improve as we increase the number of high LR training steps. The hypothesis also explains why very few (just 1) of the 50 runs trained at 0.1 LR for 30-epochs also managed to attain high accuracy – they just got lucky probabilistically and landed in a wide minimum even with a shorter duration.

To validate this hypothesis further, we run experiments similar to the one in Table 1. Specifically, we train Cifar-10 on Resnet-18 model for 200 epochs using a standard step schedule with LR of 0.1, 0.01, 0.001. We vary the number of epochs trained using the high LR of 0.1, called the *explore epochs*, from 30 to 100 epochs, and divide up the rest of the training equally between 0.01 and 0.001. For each experimental setting, we conduct 50 random trials and plot the distributions of final test accuracy and the minima sharpness as defined by the metric in Keskar et al. (2016). If our hypothesis is true, then the more you explore, the higher the probability of landing (and getting stuck) in a wide minima region, which should cause the distribution to tighten and move towards wider minima (lower sharpness), as the number of explore steps increase. This is exactly what is observed in Figure 2. Also since wide minima correlate with higher test accuracy, we should see the test accuracy distribution move towards higher accuracy and sharpen, as the number of explore steps increase. This is confirmed as well in Figure 3.

Finally, to verify whether explore at high LR is essential, we train Cifar-10 for 10,000 epochs at a fixed lower LR of 0.001. The training converged but the final test accuracy was only **93.9**. Thus, even training 50x longer at low LR is not sufficient, adding more evidence to the hypothesis.

**Multi-scale.** Given the importance of explore at high LR, a natural question that may arise is whether explore is necessary at smaller LR as well. To answer this, we train the same network for a total of 200 epochs with an initial high LR of 0.1 for 100 epochs, but now we vary the number of epochs trained with the LR of 0.01 (we call this finer-scale explore), and train with LR of 0.001 for the remaining epochs. As can be seen from Table 2, although the final training loss remains similar, we find that finer-scale explore also plays a role similar to the initial explore in determining the final test accuracy. *This indicates that our hypothesis about density of wide/narrow regions indeed holds at multiple scales*.

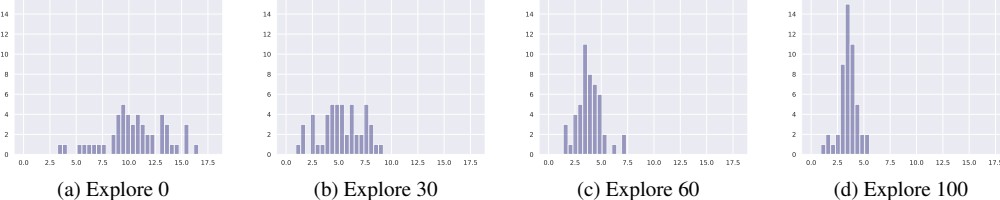

(a) Explore 0          (b) Explore 30          (c) Explore 60          (d) Explore 100

Figure 2: Histogram of minima sharpness (Keskar et al., 2016) for 50 random trials of Cifar-10 on Resnet-18. Each figure shows histograms for runs with different number of explore epochs. The distribution moves toward lower sharpness and tightens as the number of explore epochs increase.

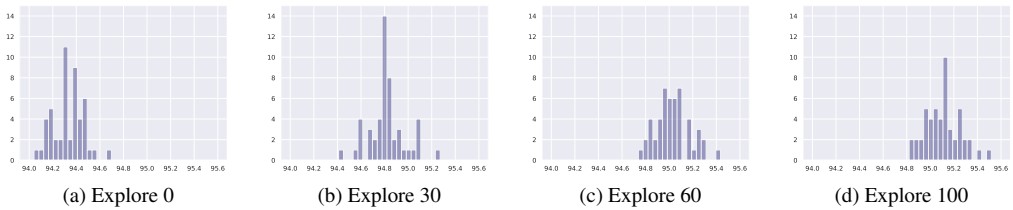

(a) Explore 0          (b) Explore 30          (c) Explore 60          (d) Explore 100

Figure 3: Histogram of test accuracy for 50 random trials of Cifar-10 on Resnet-18. Each figure shows histograms for runs with different number of explore epochs. The distribution moves toward higher test accuracy and sharpens as the number of explore epochs increase.

## 3 EXPLORE-EXPLOIT LEARNING RATE SCHEDULE

Given that we need to explore at multiple scales for good generalization, how do we go about designing a good learning rate schedule? The search space of the varying learning rate steps and their respective explore duration is enormous.

Fortunately, since the explore at the initial scale is searching over the entire loss surface while explore at finer-scales is confined to exploring only the wide-minima region identified by the initial explore, the former is more crucial. In our experiments as well, we found that the initial portion of training is much more sensitive to exploration and needs a substantial number of *explore* steps, while after this initial phase, several decay schemes worked equally well. This is similar to the observations in (Golatkar et al., 2019) where the authors found that regularization such as weight-decay and data augmentation mattered significantly only during the initial phase of training.

The above observations motivate our *Explore-Exploit* learning rate schedule, where the *explore* phase first optimizes at a high learning rate for some minimum time in order to land in the vicinity of a wide minima. We should give the *explore* phase enough time (a hyper-parameter), so that the probability of landing in a wide minima is high. After the *explore* phase, we know with a high probability, that the optimizer is in the vicinity of a wide region. We now start the *exploit* phase to descend to the bottom of this wide region while progressively decreasing the learning rate. Any smoothly decaying learning rate schedule can be thought of as doing micro *explore-exploit* at progressively reduced scales. A steady descent would allow more *explore* duration at all scales, while a fast descent would explore less at higher learning rates. We experimented with multiple schedules for the exploit phase, and found a simple linear decay to zero, that does not require any hyper-parameter, to be effective in all the models/datasets we tried. We call our proposed learning rate schedule which

Table 2: Cifar-10 on Resnet-18 trained for 200 epochs. An LR of 0.1 is used for the first 100 epochs. We then vary the number of epochs trained with LR of 0.01 (called finer-scale explore), and train the remaining epochs with an LR of 0.001. We report averages values over 3 runs.

| Explore Epochs (Finer-scale) | Test Accuracy | Training Loss | Sharpness |
|---|---|---|---|
| 10 | 94.78 | 0.0031 | 5.48 |
| 20 | 94.91 | 0.0026 | 4.47 |
| 30 | 95.00 | 0.0023 | 4.02 |
| 40 | 95.02 | 0.0021 | 3.91 |
| 50 | 95.10 | 0.0021 | 3.54 |

starts at a constant high learning rate for some minimum time, followed by a linear decay to zero, the *Knee schedule*.

Note that any learning rate decay scheme incorporates an implicit explore during the initial part, where the learning rate stays high enough. To evaluate the benefit of an explicit explore phase, we compare *Knee schedule* against several decay schemes such as linear and cosine. Interestingly, the results depend on the length of training. For long budget experiments, simple decay schemes perform comparable to *Knee schedule* in some experiments, since the implicit explore duration is also large, helping these schemes achieve good generalization. However for short budget experiments, these schemes perform significantly worse than *Knee schedule*, since the implicit explore duration is much shorter. See Table 6 and 7 for the comparison.

**Warmup.** Some optimizers such as Adam use an initial warmup phase to slowly increase the learning rate. However, as shown in Liu et al. (2019), learning rate warmup is needed mainly to reduce variance during initial training stages and can be eliminated with an optimizer such as RAdam. Learning rate warmup is also used for large-batch training (Goyal et al., 2017). Here, warmup is necessary since the learning rate is scaled to a very large value to compensate for the large batch size. This warmup is complementary and can be incorporated into *Knee schedule*. For example, we do this for $\text{BERT}_{\text{LARGE}}$ pretraining experiment where a large 16k batch size was used.

## 4 EXPERIMENTS

We have done multiple experiments in section 2 to validate our hypothesis. We do further hypothesis validation on a text dataset. This is discussed in Section A of supplementary material.

We now extensively evaluate the effectiveness of *Knee schedule* on multiple models and datasets, across various optimizers including SGD Momentum, Adam (Kingma & Ba, 2014a), RAdam (Liu et al., 2019) and LAMB (You et al., 2019). For all experiments, we used an out of the box policy, where we only change the learning rate schedule, without modifying anything else. We evaluate on image datasets – Imagenet on Resnet-50, Cifar-10 on Resnet-18; as well as NLP datasets – pre-training $\text{BERT}_{\text{LARGE}}$ on Wikipidea+BooksCorpus and fine-tuning it on SQuADv1.1; and WMT'14 (EN-DE), IWSLT'14 (DE-EN) on Transformers.

For all settings we compare *Knee schedule* against the original hand-tuned learning rate baseline for the corresponding model and dataset, showing an improvement in test accuracy in all cases. We also show that *Knee schedule* can achieve the same accuracy as the baseline with a much reduced training budget (e.g. $44\%$ less for ImageNet). Further, we also run our experiments with other common LR schedules such as linear decay, cosine decay, one-cycle (Smith, 2018). See Table 6 and Table 7 for a comparison of all LR schedules on the original budget and shorter budgets, respectively.

### 4.1 IMAGENET IMAGE CLASSIFICATION ON RESNET-50

We train ImageNet dataset (Russakovsky et al., 2015) on Resnet-50 network (He et al., 2016) with the SGD Momentum optimizer. For baseline runs, we used the standard hand-tuned step learning rate schedule of $10^{-1}, 10^{-2}, 10^{-3}$ for 30 epochs each. For *Knee schedule* we used a seed LR of 0.1 (same as baseline). We trained with the original budget of 90 epochs with an explore of 30 epochs, as well as with a reduced budget of 50 epochs. Table 3 shows the training loss and test accuracies for our experiments. *Knee schedule* comfortably beats the test accuracy of baseline in the full budget run (with absolute gains of 0.8% and 0.4% in top-1 and top-5 accuracy, respectively), while meeting the baseline accuracy even with a much shorter budget. The fact that the baseline schedule takes almost $80\%$ more training time than *Knee schedule* for the same test accuracy, shows the effectiveness of our *Explore-Exploit* scheme. See Figure 5 in Appendix for training curves.

Table 3: ImageNet on Resnet-50 results. We report mean (stddev) over 3 runs.

| LR Schedule | Test Top 1 Acc. | Test Top 5 Acc. | Training Loss | Training Epochs |
|---|---|---|---|---|
| Baseline | 75.87 (0.035) | 92.90 (0.015) | 0.74 (1e-3) | 90 |
| *Knee* | 76.71 (0.097) | 93.32 (0.031) | 0.79 (1e-3) | 90 |
| *Knee* (short budget) | 75.82 (0.080) | 92.84 (0.036) | 0.91 (0.002) | 50 |

## 4.2 CIFAR-10 IMAGE CLASSIFICATION ON RESNET-18

We train Cifar-10 dataset (Krizhevsky et al., 2009) on Resnet-18 network (He et al., 2016) with the SGD Momentum optimizer. For baseline, we used the hand-tuned step learning rate schedule of $10^{-1}, 10^{-2}, 10^{-3}$ for 100, 50, 50 epochs, respectively. With *Knee schedule*, we train the network with the original budget of 200 epochs, as well as a reduced budget of 150 epochs. We used 100 explore epochs for both runs, and a seed learning rate of 0.1 (same as baseline). Table 6 and Table 7 shows the test accuracy for the full and reduced budget runs. *Knee schedule* beats the test accuracy of baseline in the full budget run, while meeting the baseline test accuracy in 25% less budget. See Table 10, figure 6 in Appendix for detailed comparisons of training loss, test accuracy, and LR.

## 4.3 BERT$_{\text{LARGE}}$ PRE-TRAINING

We now evaluate *Knee schedule* on a few NLP tasks. In the first task, we pre-train the BERT$_{\text{LARGE}}$ model (Devlin et al., 2018) on the Wikipedia+BooksCorpus dataset with LAMB (You et al., 2019) optimizer using a batch-size of 16k.

Since large batch training requires learning rate warmup (see Goyal et al. (2017)), we incorporate it into the *Knee schedule* by first doing a warmup followed by the explore-exploit phases. We used an explore of 50% steps for both phases of BERT training. For baseline, we use the warmup + linear decay schedule (You et al., 2019; Devlin et al., 2018). The pre-trained models are evaluated on the SQuAD v1.1 (Rajpurkar et al., 2016) dataset by fine-tuning on the dataset for 2 epochs. We pretrain the model on both the full budget, as well as with 33% less budget. See Table 4 for the results. For the full budget run, *Knee schedule* improves the baseline by 0.2%, while for the reduced budget we achieved similar fine-tuning accuracy as baseline. The baseline schedule achieves a much lower accuracy with shorter budget training, showing the efficacy of *Knee schedule*. BERT pre-training is extremely compute expensive and takes around 47 hours on 64 V100 GPUs (3008 V100 GPU-hrs) on cloud VMs. The reduced budget amounts to a saving of approximately 1002 V100 GPU-hours!

Table 4: BERT$_{\text{LARGE}}$ results. We report the pre-training train loss, and the test F1 accuracy on SQuAD v1.1 after fine-tuning. See figure 7 in Appendix for training curves.

| LR Schedule | F1 score on SQuAD v1.1 | Training loss | Total Training Steps |
|---|---|---|---|
| Knee | 91.51 | 1.248 | 31250 |
| Baseline (You et al., 2019) | 91.34 | - | 31250 |
| Baseline (short budget) | 90.64 | 1.336 | 20854 |
| *Knee* (short budget) | 91.29 | 1.275 | 20854 |

## 4.4 MACHINE TRANSLATION ON TRANSFORMER NETWORK WITH WMT'14 AND IWSLT

In the second NLP task, we train the Transformer (base model) (Vaswani et al., 2017) on the IWSLT'14 (De-En) (Cettolo et al., 2014) and WMT'14 (En-De) (Bojar et al., 2014) datasets with the RAdam (Liu et al., 2019) optimizer.

**WMT'14 (EN-DE):** The baseline schedule uses a linear decay for 70 epochs (Liu et al., 2019). With *Knee schedule*, we trained with the original budget of 70 epochs, as well as a reduced budget of 30 epochs. We used 50 and 25 explore epochs for the two runs, respectively and a seed learning rate of $3e - 4$ for both *Knee schedule* and baseline. In all cases we use the model checkpoint with least loss on the validation set for computing BLEU scores on the test set. Table 5 shows the training loss and test accuracy averaged over 3 runs. *Knee schedule* improves the test BLEU score of baseline in the full budget run by 0.24 points. In the shorter budget run, *Knee schedule* matches the test accuracy of the baseline while taking 57% less training time (a saving of 80 V100 GPU-hours!)

**IWSLT'14 (DE-EN):** The baseline schedule uses a linear decay for 50 epochs (Liu et al., 2019). With *Knee schedule*, we trained with the original budget of 50 epochs, as well as a reduced budget of 35 epochs. We used 40 and 30 explore epochs for the two runs, respectively and a seed LR of $3e - 4$ for both *Knee schedule* and baseline. In all cases we use the model checkpoint with least loss on the validation set for computing BLEU scores on the test set. Table 6 and 7 show the average test accuracy for full and reduced budget runs. *Knee schedule* improves the baseline test BLEU score by

Table 5: Results for WMT'14 (EN-DE) on Transformer networks. The test BLEU scores are computed on the checkpoint with the best validation perplexity. We report mean (stdev) over 3 runs.

| LR Schedule | Test BLEU Score | Train Perplexity | Validation Perplexity | Training Epochs |
|---|---|---|---|---|
| Baseline | 27.29 (0.06) | 3.87 (0.017) | 4.89 (0.02) | 70 |
| *Knee* | 27.53 (0.12) | 3.89 (0.017) | 4.87 (0.006) | 70 |
| *Knee* (short budget) | 27.28 (0.17) | 4.31 (0.02) | 4.92 (0.007) | 30 |

0.56 points in the full budget run. In the shorter budget run, *Knee schedule* matches the test accuracy of the baseline schedule while taking 30% less training time. Also see Table 11 in Appendix. See Figure 8 and Figure 9 in Appendix for training curves.

## 4.5 State of the Art Results

To further demonstrate the effectiveness of *Knee schedule*, we took a recent high performing model, MAT (Fan et al., 2020), which had shown very high accuracy on the IWSLT'14 (DE-EN) dataset with a BLEU score of 36.22 trained with inverse square root LR schedule. We simply retrained the model with *Knee schedule*, and achieved a new SOTA BLEU score of 36.6 (a 0.38 point increase). We also trained the model with WMT'14 (DE-EN) dataset and again achieved a SOTA BLEU score of 31.9 with *Knee schedule*, compared to the baseline score of 31.34 which uses the inverse square root LR schedule. See Section B.5, Table 12, Table 13 and Figure 10 in Appendix for more details.

## 4.6 Comparison with other learning schedules

We also ran all our experiments with multiple other learning rate schedules – one-cycle (Smith, 2018), cosine decay (Loshchilov & Hutter, 2016) and linear decay. See section E in supplementary material for details of these learning rate schedules, and a detailed performance comparison. Note that in *Knee schedule*, the explore duration is a hyperparameter. To avoid tuning this hyperparameter, we also experimented with a fixed 50% explore duration for the full budget runs. Table 6 shows the test accuracies of the various experiments on all schedules, when trained with the original budget; while Table 7 shows the results when trained with a reduced budget. As shown, for the full budget runs, *Knee schedule* improves on the test accuracies on all experiments. Even the fixed 50% explore *Knee schedule* outperforms all the other baselines. Also noteworthy is that *Knee schedule* is able to achieve the same test accuracies as the baseline's full budget runs with a much lower training budget, saving precious GPU cycles. We also demonstrate that the delta in accuracy between *Knee schedule* and other decay schemes is non-trivial by computing the number of epochs needed by each scheme to reach the target BLEU scores for WMT'14 and IWSLT'14. As shown in Table 8, *Knee schedule* is highly efficient compared to say Cosine Decay which takes 100% more training time to achieve the same accuracy for WMT'14.

Table 6: We report the top-1 accuracy for ImageNet and Cifar-10, BLEU score for IWSLT'14 and WMT'14 and F1 score for BERT on SQuAD. All values are averaged over multiple runs.

| Experiment | Training Budget (epochs) | *Knee Schedule* | *Knee Schedule* (Fixed 50% explore) | Baseline | One-Cycle | Cosine Decay | Linear Decay |
|---|---|---|---|---|---|---|---|
| ImageNet | 90 | **76.71** | 76.58 | 75.87 | 75.39 | 76.41 | 76.54 |
| Cifar-10 | 200 | **95.26** | 95.26 | 95.10 | 94.09 | 95.23 | 95.18 |
| IWSLT | 50 | **35.53** | 35.23 | 34.97 | 34.77 | 35.21 | 34.97 |
| WMT'14 | 70 | **27.53** | 27.41 | 27.29 | 27.19 | 27.35 | 27.29 |
| BERT$_{\text{LARGE}}$ | 31250 (iters) | **91.51** | 91.51 | 91.34 | - | - | 91.34 |

## 5 Related Work

**Generalization.** There has been a lot of work on understanding the generalization characteristics of DNNs. Kawaguchi (2016) found that DNNs have many local minima, but all local minima were also the global minima. It has been observed by several authors that wide minima generalize

Table 7: Shorter budget training: Test accuracy on all learning rate schedules tried in this paper, but trained with a shortened budget. We report same metrics as Table 6. *Knee schedule* achieves the same accuracy as baseline schedules in much lower budget, saving precious GPU-hours.

| Experiment | Shortened Training Budget (epochs) | *Knee Schedule* | One-Cycle | Cosine Decay | Linear Decay | Saving ( V100 GPU-hours) |
|---|---|---|---|---|---|---|
| ImageNet | 50 | **75.82** | 75.36 | 75.71 | 75.82 | 27 |
| Cifar-10 | 150 | **95.14** | 93.84 | 95.06 | 95.02 | 0.25 |
| IWSLT | 35 | **35.08** | 34.43 | 34.46 | 34.16 | 0.75 |
| WMT'14 | 30 | **27.28** | 26.80 | 26.95 | 26.77 | 80 |
| BERT$_{LARGE}$ | 20854 (iterations) | **91.29** | - | - | 90.64 | 1002 |

Table 8: Epochs required by different LR schedules to reach the target accuracy.

| Experiment | Target BLEU Score | *Knee schedule* | Cosine Decay | Linear Decay |
|---|---|---|---|---|
| IWSLT | 35.08 | 35 | 45 | 60 |
| WMT'14 | 27.28 | 30 | 60 | 70 |

better than narrow minima (Arora et al., 2018; Hochreiter & Schmidhuber, 1997; Keskar et al., 2016; Jastrzebski et al., 2017; Wang et al., 2018) but there have been other works questioning this hypothesis as well (Dinh et al., 2017; Golatkar et al., 2019; Guiroy et al., 2019; Jastrzebski et al., 2019; Yoshida & Miyato, 2017).

Keskar et al. (2016) found that small batch SGD generalizes better and lands in wider minima than large batch SGD. However, recent work has been able to generalize quite well even with very large batch sizes (Goyal et al., 2017; McCandlish et al., 2018; Shallue et al., 2018), by scaling the learning rate linearly as a function of the batch size. Jastrzebski et al. (2019) analyze how batch size and learning rate influence the curvature of not only the SGD endpoint but also the whole trajectory. They found that small batch or large step SGD have similar characteristics, and yield smaller and earlier peak of spectral norm as well as smaller largest eigenvalue. Chaudhari et al. (2019); Baldassi et al. (2019) propose methods to drive the optimizer to wide minima. Wang et al. (2018) analytically show that generalization of a model is related to the Hessian and propose a new metric for the generalization capability of a model that is unaffected by model reparameterization of Dinh et al. (2017). Yoshida & Miyato (2017) argue that regularizing the spectral norm of the weights of the neural network help them generalize better. On the other hand, Arora et al. (2018) derive generalization bounds by showing that networks with low stable rank (high spectral norm) generalize better. Guiroy et al. (2019) looks at generalization in gradient-based meta-learning and they show experimentally that generalization and wide minima are not always correlated.

**Lower density of wide minima.** Wu et al. (2018) compares the sharpness of minima obtained by full-batch gradient descent (GD) with different learning rates for small neural networks on Fashion-MNIST and Cifar10 datasets. They find that GD with a given learning rate finds the theoretically sharpest feasible minima for that learning rate. Thus, in the presence of several flatter minimas, GD with lower learning rates does not find them, leading to the conjecture that density of sharper minima is perhaps larger than density of wider minima.

## 6 CONCLUSIONS

In this paper, we make an observation that an initial *explore* phase with a high learning rate is essential for good generalization of DNNs. Further, we find that a minimum *explore* duration is required even if the training loss stops improving much earlier. We explain this observation via our hypothesis that in the DNN loss landscape, the density of wide minima is significantly lower than that of narrow minima. Motivated by this hypothesis, we present an *Explore-Exploit* based learning rate schedule, called the *Knee schedule*. We do extensive evaluation of *Knee schedule* on multiple models and datasets. In all experiments, the *Knee schedule* outperforms prior hand-tuned baselines, including achieving SOTA test accuracies, when trained with the original training budget, and achieves the same test accuracy as the baseline when trained with a much shorter budget.

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

## A    HYPOTHESIS VALIDATION

For validating our hypothesis on the density of wide minima vs narrow minima, we did multiple experiments, most of which were discussed in section 2 of the main paper. To summarize, in figures 2, 3 of the main paper, we showed that for Cifar-10 on Resnet-18, as the number of explore steps increase, the distribution of minima width and test accuracy sharpens and shifts towards wider minima and better accuracy, respectively. This behaviour is predictable from our hypothesis as increasing explore steps increases the probability of landing in a wide region. From the same argument, the average accuracy should increase as the number of explore steps increase, which is confirmed in Table 1 of the main paper. Our hypothesis also predicts that even at low explore epochs, although the probability of landing in a wide region is low, it is non zero. Thus, out of many trials with low number of explore epochs, a few runs could still yield high test accuracy. This is what we observe in Figure 3(b) of the main paper, where 1 out of 50 trials for the 30 explore runs obtains an accuracy more than 95.24 even though the average accuracy for 30 explore is 94.81!.

Table 9: IWSLT'14 (DE-EN) on the Transformer network trained with the *Knee schedule*. The *explore* duration is varied, while keeping the total training budget fixed at 50 epochs. We report averages over 3 runs.

| Explore Epochs | Test BLEU score | Training Perplexity |
|---|---|---|
| 5 | 34.93 | 3.29 |
| 10 | 35.02 | 3.22 |
| 15 | 35.08 | 3.11 |
| 20 | 35.10 | 3.08 |
| 25 | 35.23 | 3.02 |
| 30 | 35.28 | 2.99 |
| 40 | 35.53 | 3.00 |

We do a similar experiments on the IWSLT'14 German to English dataset (Cettolo et al., 2014) trained on Transformer networks (Vaswani et al., 2017) to demonstrate that our hypothesis holds even on a completely different NLP dataset and network architecture. We train with the *Knee schedule* for a total budget of 50 epochs, but keep varying the number of explore epochs. As shown in Table 9, the test BLEU score increases as we increase the number of explore epochs. Further, we found that among multiple trials, a 20 epoch explore run had a high BLEU score of 35.29, suggesting that the run got lucky. Thus, these results on the IWSLT'14 (DE-EN) dataset add more evidence to the wide-minima density hypothesis.

Also, see section D and Table 17 for a sensitivity analysis of the *Knee schedule* on the starting learning rate. Interestingly, we found that the optimal explore duration varies inversely with the starting LR. Since a bigger learning rate has higher probability of escaping narrow minima compared to a lower learning rate, it would, on an average, require fewer steps to land in a wide minima. Thus, larger learning rates can explore faster. This observation is thus consistent with our hypothesis and further corroborates it.

## B    EXPERIMENT DETAILS

In this section we describe the implementation and other details of each experiment. For the smaller datasets – Cifar-10 and IWSLT'14 (DE-EN), we also provide the detailed training loss, test accuracy table which we skipped in the main paper. We also discuss an additional experiment, where we fine-tune a pretrained $BERT_{BASE}$ network on the SQuAD-v1.1 dataset.

### B.1    IMAGENET IMAGE CLASSIFICATION ON RESNET-50

We train ImageNet Russakovsky et al. (2015) on Resnet-50 He et al. (2016), which has 25 million parameters, with a batch size of 256 and a seed learning rate of 0.1. Random cropping and random horizontal flipping augmentations were applied to the training dataset. We use SGD optimizer with

momentum of 0.9 and weight decay of 1e-4 [1]. For knee schedule, we choose explore as 30 epochs. Figure 5 shows the training loss, test accuracy and learning rate curves during training.

## B.2 CIFAR-10 IMAGE CLASSIFICATION ON RESNET-18

For this experiment, we use a batch size of 128 and a seed learning rate of 0.1 with ResNet-18He et al. (2016) which has around 11 million parameters. SGD optimizer is used with momentum of 0.9 and weight decay of 5e-4 [2]. Random cropping and random horizontal flipping augmentations were applied to the training dataset. For knee schedule, we choose 100 epochs as explore for short and full budget runs. Table 10 shows the training loss and test accuracies for the various runs. Figure 6 shows the training loss, test accuracy and learning rate curves during training.

Table 10: Training loss and Test accuracy for Cifar-10 on Resnet-18. We report mean (stddev) over 7 runs.

| LR Schedule | Test Accuracy | Training Loss | Training Epochs |
|---|---|---|---|
| Baseline | 95.10 (0.14) | 0.002 (1e-4) | 200 epochs |
| *Knee* | 95.26 (0.11) | 0.002 (1e-4) | 200 epochs |
| *Knee* (short budget) | 95.14 (0.18) | 0.004 (3e-4) | 150 epochs |

## B.3 BERT$_{LARGE}$ PRE-TRAINING

We pre-train BERT using the LAMB optimizer [3]. BERT$_{LARGE}$ has around 330 million parameters and the pre-training is divided into two phases with different sequence lengths. The first phase consists of 90% steps with sequence length of 128 and the second phase consists of the remaining 10% steps with sequence length of 512 Devlin et al. (2018). We used a batch size of 16384 in both phases of training. We train the model on a shortened training budget of 2/3rd the original steps (20854 instead of 31250 steps) and on the full budget (31250 steps). For *Knee schedule*, we explore for 50% of the phase budget after performing warmup for 10% of the phase steps. Please refer to Figure 7 for the training loss and learning rate curves of the short budget runs.

## B.4 MACHINE TRANSLATION ON TRANSFORMER NETWORK WITH WMT'14(EN-DE) AND IWSLT'14(DE-EN)

We use the default implementation provided by the fairseq package Ott et al. (2019) [4]. We train WMT'14 (EN-DE) dataset on the Transformer$_{BASE}$ Vaswani et al. (2017) model which has around 86 million parameters and use the RAdam Liu et al. (2019) optimizer with $\beta_1$ of 0.9 and $\beta_2$ of 0.999. Label smoothed cross entropy was used as the objective function with an uncertainty of 0.1. A dropout of 0.1, clipping norm of 25 and weight decay of 1e-4 is used. Each training batch contains approximately 30000 tokens. For *Knee schedule*, we choose explore of 25 epochs for short budget runs and 50 epochs for full budget runs. Please see Figure 8 for training perplexity, validation perplexity and learning rate curves.

For IWSLT'14 we use the same configuration as WMT'14, except for a dropout of 0.3 following Fairseq's out-of-box implementation. Each training batch contains approximately 4000 tokens. For *Knee schedule*, we choose explore as 30 epochs for short budget runs and 40 epochs for full budget runs. Table 11 shows the training loss and test accuracies for the various runs. Figure 9 shows the training perplexity, validation perplexity and learning rate curves.

---

[1] We used the opensource implementation at: https://github.com/cybertronai/imagenet18_old

[2] We used the open-source implementation at: https://github.com/kuangliu/pytorch-cifar

[3] We used the open-source implementation at: https://github.com/NVIDIA/DeepLearningExamples/tree/master/PyTorch/LanguageModeling/BERT

[4] https://github.com/pytorch/fairseq

Table 11: Training, validation perplexity and test BLEU scores for IWSLT on Transformer networks. The test BLEU scores are computed on the checkpoint with the best validation perplexity. We report the mean and standard deviation over 3 runs.

| LR Schedule | Test BLEU Score | Train Perplexity | Validation Perplexity | Training Epochs |
|---|---|---|---|---|
| Baseline | 34.97 (0.035) | 3.36 (0.001) | 4.91 (0.035) | 50 |
| *Knee* | 35.53 (0.06) | 3.00 (0.044) | 4.86 (0.02) | 50 |
| *Knee* (short budget) | 35.08 (0.12) | 3.58 (0.049) | 4.90 (0.063) | 35 |

## B.5 State of the Art Results using the MAT model with IWSLT'14 (DE-EN) and WMT'14 (DE-EN) datasets

We use the default implementation provided by the authors of Fan et al. (2020) [5]. The MAT model is trained in two phases. A smaller model is trained in the first phase, which is used to initialize a larger model of the second phase. Adam optimizer with $\beta_1$ of 0.9 and $\beta_2$ of 0.98 is used for both phases.

**IWSLT'14 (DE-EN):** For the baseline run with IWSLT'14 (DE-EN) dataset, we follow the exact instructions provided by the authors in their code repository – each phase is trained for 200 epochs, using an inverse square root LR schedule with warmup of 4000 steps and a peak LR of $5e^{-4}$. For the run with *Knee schedule* we simply replace the learning rate schedule, keeping everything else unchanged. We used the same warmup of 4000 steps, and explored at $5e^{-4}$ LR till 100 epochs followed by the linear decay exploit phase. Table 12 shows the train loss and test accuracies for the various runs. Please see Figure 10 for detailed plots.

Table 12: Training, validation perplexity and test BLEU scores for IWSLT'14 DE-EN on MAT. The test BLEU scores are computed on the checkpoint with the best validation perplexity. Results are averaged over 3 runs.

| LR Schedule | Test BLEU Score | Train Perplexity | Validation Perplexity | Training Epochs |
|---|---|---|---|---|
| Baseline (inv. sqrt) | 36.20 (0.02) | 3.87 ( 0.009) | 4.58 (0.009) | 200 + 200 (each phase) |
| *Knee schedule* | 36.59 (0.017) | 3.74 (0.012) | 4.45 ( 0.009) | 200 + 200 (each phase) |

**WMT'14 (DE-EN):** We follow the instructions provided in Fan et al. (2020) to preprocess the train and test datasets. For baseline, we train each phase for 70 epochs, using an inverse square root LR schedule with warmup of 4000 steps and a peak LR of $5e^{-4}$. For the run with *Knee schedule* we simply replace the learning rate schedule, keeping everything else unchanged. We used the same warmup of 4000 steps, and explored at $5e^{-4}$ LR for 35 epochs followed by the linear decay exploit phase. Table 13 shows the train loss and test accuracies for the various runs.

Table 13: Training, validation perplexity and test BLEU scores for WMT'14 DE-EN on MAT. The test BLEU scores are computed on the checkpoint with the best validation perplexity.

| LR Schedule | Test BLEU Score | Train Perplexity | Validation Perplexity | Training Epochs |
|---|---|---|---|---|
| Baseline (inv. sqrt) | 31.34 | 4.78 | 4.75 | 70 + 70 (each phase) |
| *Knee schedule* | 31.90 | 4.67 | 4.68 | 70 + 70 (each phase) |

---

[5]https://github.com/HA-Transformer/HA-Transformer

### B.6 SQuAD-v1.1 FINE-TUNING ON BERT$_{\text{BASE}}$

We also evaluate *Knee schedule* on the task of fine-tuning BERT$_{\text{BASE}}$ model Devlin et al. (2018) on SQuAD v1.1 Rajpurkar et al. (2016) with the Adam Kingma & Ba (2014b) optimizer [6]. BERT fine-tuning is prone to overfitting because of the huge model size compared to the small fine-tuning dataset, and is typically run for only a few epochs. For baseline we use the linear decay schedule mentioned in Devlin et al. (2018). We use a seed learning rate of $3e^{-5}$ and train for 2 epochs. For *Knee schedule*, we train the network with 1 explore epoch with the same seed learning rate of $3e^{-5}$. Table 14 shows our results over 3 runs. We achieve a mean EM score of 81.4, compared to baseline's 80.9, a 0.5% absolute improvement. We don't do a short budget run for this example, as the full budget is just 2 epochs. Please refer to Figure 11 for the training loss, test accuracy and learning rate curves.

Table 14: SQuAD fine-tuning on BERT$_{\text{BASE}}$. We report the average training loss, and average test EM, F1 scores over 3 runs.

| LR Schedule | EM | F1 | Train Loss | Training Epochs |
|---|---|---|---|---|
| Baseline | 80.89 (0.15) | 88.38 (0.032) | 1.0003 (0.004) | 2 |
| *Knee schedule* | 81.38 (0.02) | 88.66 (0.045) | 1.003 (0.002) | 2 |

## C  MINIMA SHARPNESS

Our hypothesis predicts that a higher explore should help the optimizer in finding a wider minimum, which in turn helps generalization. For quantitative evaluation, we used two different metrics for measuring the minima width, and evaluate the effect of explore on the width of the converged minima.

### C.1  KESKAR'S SHARPNESS METRIC KESKAR ET AL. (2016)

In our first evaluation we used the sharpness metric proposed in Keskar et al. (2016) (see Metric 2.1 in section 2.2.2 of the paper). This metric was used to compute the histograms in Figure 2 of the main paper as well. We use an $\epsilon$ of 1e-4, and $A$ is chosen to be the identity matrix. The maximization problem is solved by applying 1000 iterations of projected gradient ascent. We compute this metric for the Cifar-10 on Resnet-18 experiments for different number of explore epochs. As shown in Table 15, the average sharpness of the minima decreases as we increase the number of explore epochs, as predicted by our hypothesis.

Table 15: Keskar's sharpness metric for Cifar-10 on Resnet-18 trained for 200 epochs with Momentum. An LR of 0.1 is used for the explore epochs. Half the remaining epochs are trained at 0.01 and the other half at 0.001. We report the average sharpness over 50 different trials.

| Explore Epochs | Sharpness |
|---|---|
| 0 | 10.56 |
| 30 | 5.43 |
| 60 | 3.86 |
| 100 | 3.54 |

### C.2  FISHER SCORE

The maximum Eigen value of the Fisher Information Matrix (FIM) is another metric used to measure the maximum curvature of the loss landscape (see Sokol & Park (2018)). We used an unbiased estimate of the true Fisher matrix (see Kunstner et al. (2019)) using 10 unbiased samples per training data. Table 16 shows the average FIM scores for the Cifar-10 experiments with different explores. As shown, the FIM score decreases as the number of explore epochs increases, indicating that higher

---

[6] We used the implementation at: https://github.com/huggingface/pytorch-transformers

explore leads to wider minimum with a lower curvature. This is in line with the prediction from our hypothesis.

Table 16: FIM Score for Cifar-10 on Resnet-18 trained for 200 epochs with Momentum. A LR of 0.1 is used for the explore epochs. Half the remaining epochs are trained at 0.01 and the other half at 0.001. We report the average FIM score over 10 different trials.

| Explore Epochs | FIM score |
|---|---|
| 0 | 0.051 |
| 30 | 0.046 |
| 60 | 0.043 |
| 100 | 0.042 |

## D  LEARNING RATE SENSITIVITY

We performed sensitivity analysis of the starting learning rate, referred to as the seed learning rate, for *Knee schedule*. We trained the Cifar-10 dataset on Resnet-18 with the *Knee schedule* for a shortened budget of 150 epochs, starting at different seed LRs. For each experiment, we do a simple linear search to find the best explore duration. The test accuracies and optimal explore duration for the different seed LR choices is shown in Table 17. As shown, the seed learning rate can impact the final accuracy, but *Knee schedule* is not highly sensitive to it. In fact, we can achieve the target accuracy of 95.1 with multiple seed learning rates of 0.05, 0.075, 0.0875 and 0.115, as compared to the original seed learning rate of 0.1, by tuning the number of explore epochs.

Another interesting observation is that the optimal explore duration varies inversely with the seed LR. Since a bigger learning rate has higher probability of escaping narrow minima compared to a lower learning rate, it would, on an average, require fewer steps to land in a wide minima. Thus, larger learning rates can *explore* faster, and spend more time in the *exploit* phase to go deeper in the wide minimum. This observation is thus consistent with our hypothesis and further corroborates it.

We also note that by tuning both seed LR and explore duration, we can achieve the twin objectives of achieving a higher accuracy, as well as a shorter training time – e.g. here we are able to achieve an accuracy of 95.34 in 150 epochs (seed LR 0.075), compared to 95.1 achieved by the baseline schedule in 200 epochs.

Table 17: Seed LR sensitivity analysis. Cifar-10 on Resnet-18 trained for 150 epochs with *Knee schedule*. We vary the seed LR and explore epochs to get the best test accuracy for the particular setting. We report averages over 3 runs.

| Seed LR | Test Accuracy | Optimal Explore Epochs |
|---|---|---|
| 0.03 | 95.07 | 120 |
| 0.05 | 95.12 | 120 |
| 0.0625 | 95.15 | 120 |
| 0.075 | 95.34 | 100 |
| 0.0875 | 95.22 | 100 |
| 0.1 | 95.14 | 100 |
| 0.115 | 95.20 | 60 |
| 0.125 | 95.06 | 60 |
| 0.15 | 95.04 | 30 |

## E  COMPARISONS WITH MORE BASELINE LEARNING RATE SCHEDULES

In this section we compare *Knee schedule* against more learning rate schedules – one-cycle, linear decay and cosine decay.

**One-Cycle**: The one-cycle learning rate schedule was proposed in Smith (2018) (also see Smith (2017)). This schedule first chooses a maximum learning rate based on an LR Range test. The LR

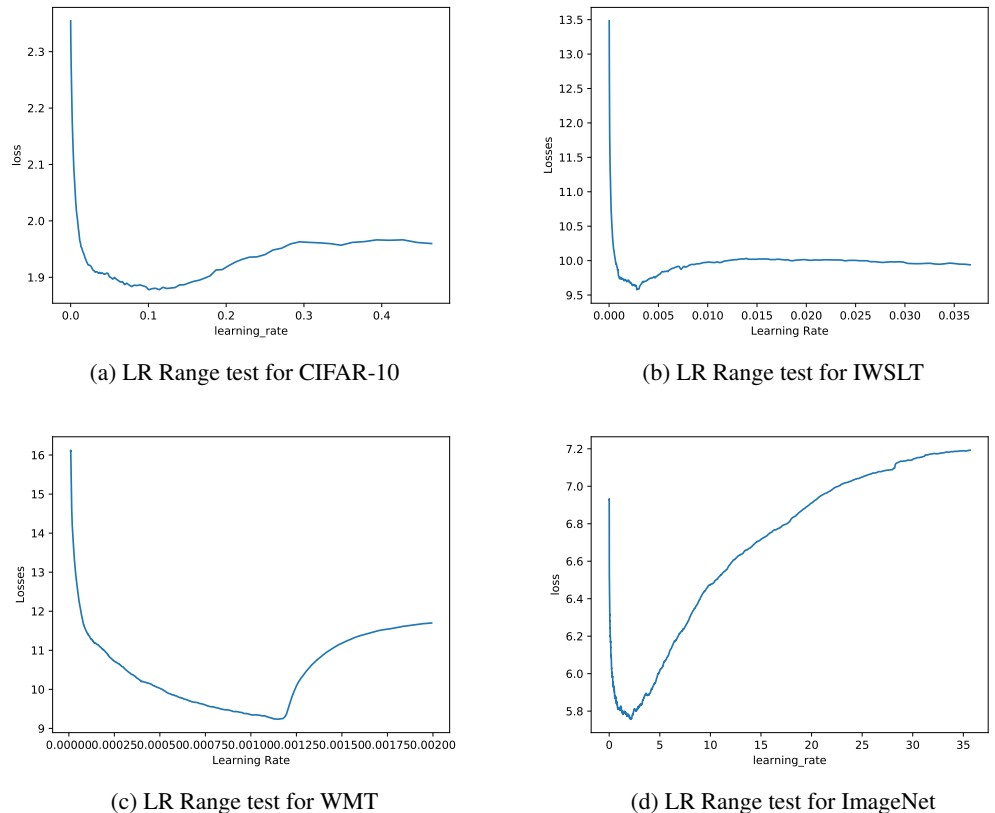

Figure 4: LR Range test for selecting the maximum learning rate. A good choice is the learning rate is a bit before the minima in a region where the loss is still decreasing.

range test starts from a small learning rate and keeps increasing the learning rate until the loss starts exploding (see figure 4). Smith (2018) suggests that the maximum learning rate should be chosen to be bit before the minima, in a region where the loss is still decreasing. There is some subjectivity in making this choice, although some blogs and libraries[7] suggest using a learning rate one order lower than the one at minima. We go with this choice for all our runs.

Once the maximum learning rate is chosen, the one-cycle schedule proceeds as follows. The learning rate starts at a specified fraction[8] of the maximum learning rate and is increased linearly to the maximum learning rate for 45 percent of the training budget and then decreased linearly for the remaining 45. For the final 10 percent, the learning rate is reduced by a large factor (we chose a factor of 10). We used an opensource implementation [9] for our experiments.

**Linear Decay**: The linear decay learning rate schedule simply decays the learning rate linearly to zero starting from a seed LR.

**Cosine Decay**: The cosine decay learning rate schedule decays the learning rate to zero following a cosine curve, starting from a seed LR.

---

[7]See e.g. https://towardsdatascience.com/finding-good-learning-rate-and-the-one-cycle-policy-7 and https://sgugger.github.io/how-do-you-find-a-good-learning-rate.html. Also see https://docs.fast.ai/callbacks.lr_finder.html and https://docs.fast.ai/callbacks.one_cycle.html

[8]See div_factor in https://docs.fast.ai/callbacks.one_cycle.html. We chose the fraction to be 0.1 in our experiments.

[9]https://github.com/nachiket273/One_Cycle_Policy

### E.1 CIFAR-10

Figure 4a shows the LR range test for Cifar-10 with the Resnet-18 network. The minima occurs around learning rate of 0.09, and we choose $9e-3$ as the maximum learning rate for the One-Cycle runs. For linear, cosine decay schedules we start with a seed learning rate of 0.1 as used in the standard baselines. The training loss and test accuracy for the various schedules are shown in Table 18 for the full budget runs (200 epochs), and in Table 19 for the short budget runs (150 epochs).

Table 18: Cifar-10 on Resnet-18 full budget training (200 epochs): Training loss and Test accuracy for more learning rate schedules. We report the mean and standard deviation over 7 runs.

| LR Schedule | Test Accuracy | Train Loss |
|---|---|---|
| One-Cycle | 94.08 (0.07) | 0.0041 (6e-5) |
| Cosine Decay | 95.23 (0.11) | 0.0023 (9e-5) |
| Linear Decay | 95.18 (0.15) | 0.0018 (7e-5) |
| *Knee schedule* | **95.26** (0.11) | 0.0023 (1e-4) |

Table 19: Cifar-10 on Resnet-18 short budget training (150 epochs): Training loss and Test accuracy for more learning rate schedules. We report the mean and standard deviation over 7 runs.

| LR Schedule | Test Accuracy | Train Loss |
|---|---|---|
| One-Cycle | 93.84 (0.082) | 0.0052 (7e-5) |
| Cosine Decay | 95.06 (0.16) | 0.0030 (2e-4) |
| Linear Decay | 95.02 (0.10) | 0.0021 (1e-4) |
| *Knee schedule* | **95.14** (0.18) | 0.0044 (3e-4) |

### E.2 IMAGENET

Figure 4d shows the LR range test for ImageNet with the Resnet-50 network. The minima occurs around learning rate of 2.16, and we choose $0.216$ as the maximum learning rate for One-Cycle runs. For linear, cosine decay schedules we start with a seed learning rate of 0.1 as used in the standard baselines. The training loss and test accuracy for the various schedules are shown in Table 20 for the full budget runs (90 epochs), and in Table 21 for the short budget runs (50 epochs).

Table 20: ImageNet with ResNet-50 full budget training (90 epochs): Training loss, Test Top-1 and Test Top-5 for more learning rate schedules. We report the mean and standard deviation over 3 runs.

| LR Schedule | Test Top-1 | Test Top-5 | Train Loss (av) |
|---|---|---|---|
| One Cycle | 75.39 (0.137) | 92.56 (0.040) | 0.96 (0.003) |
| Cosine Decay | 76.41 (0.212) | 93.28 (0.066) | 0.80 (0.002) |
| Linear decay | 76.54 (0.155) | 93.21 (0.051) | 0.75 (0.001) |
| *Knee schedule* | **76.71** (0.097) | **93.32** (0.031) | 0.79 (0.001) |

Table 21: ImageNet with ResNet-50 short budget training (50 epochs): Training loss, Test Top-1 and Test Top-5 for more learning rate schedules. We report the mean and standard deviation over 3 runs.

| LR Schedule | Test Top-1 | Test Top-5 | Train Loss (av) |
|---|---|---|---|
| One Cycle | 75.36 (0.096) | 92.53 (0.079) | 1.033 (0.004) |
| Cosine Decay | 75.71 (0.116) | 92.81 (0.033) | 0.96 (0.002) |
| Linear decay | 75.82 (0.080) | 92.84 (0.036) | 0.91 (0.002) |
| *Knee schedule* | **75.82** (0.080) | **92.84** (0.036) | 0.91 (0.002) |

### E.3 WMT'14 EN-DE

Figure 4c shows the LR range test for WMT'14 EN-DE on the transformer networks. The minima occurs near 1.25e-3. For the maximum learning rate, we choose 2.5e-4 for the default one-cycle policy. For linear, cosine decay schedules we start with a seed learning rate of 3e-4 as used in the standard baselines The training, validation perplexity and BLEU scores for the various schedules are shown in Table 22 for the full budget runs (70 epochs), and in Table 23 for the short budget runs (30 epochs).

Table 22: WMT'14 (EN-DE) on Transformer networks full budget training (70 epochs): Training, validation perplexity and test BLEU scores for more learning rate schedules. The test BLEU scores are computed on the checkpoint with the best validation perplexity. We report the mean and standard deviation over 3 runs.

| LR Schedule | Test BLEU Score | Train ppl | Validation ppl |
|---|---|---|---|
| One-Cycle | 27.19 (0.081) | 3.96 (0.014) | 4.95 (0.013) |
| Cosine Decay | 27.35 (0.09) | 3.87 (0.011) | 4.91 (0.008) |
| Linear Decay | 27.29 (0.06) | 3.87 (0.017) | 4.89 (0.02) |
| *Knee schedule* | **27.53** (0.12) | 3.89 (0.017) | 4.87 (0.006) |

Table 23: WMT'14 (EN-DE) on Transformer networks short budget training (30 epochs): Training, validation perplexity and test BLEU scores for more learning rate schedules. The test BLEU scores are computed on the checkpoint with the best validation perplexity. We report the mean and standard deviation over 3 runs.

| LR Schedule | Test BLEU Score | Train ppl | Validation ppl |
|---|---|---|---|
| One-Cycle | 26.80 (0.2) | 4.38 (0.017) | 5.02 (0.007) |
| Cosine Decay | 26.95 (0.23) | 4.32 (0.013) | 4.99 (0.011) |
| Linear Decay | 26.77 (0.12) | 4.36 (0.092) | 5.02 (0.01) |
| *Knee schedule* | **27.28** (0.17) | 4.31 (0.02) | 4.92 (0.007) |

### E.4 IWSLT'14 DE-EN

Figure 4b shows the LR range test for IWSLT on the transformer networks. The minima occurs near 2.5e-3. For the maximum learning rate, we choose 2.5e-4 for the default one-cycle policy. For linear, cosine decay schedules we start with a seed learning rate of 3e-4 as used in the standard baselines The training, validation perplexity and BLEU scores for the various schedules are shown in Table 24 for the full budget runs (50 epochs), and in Table 25 for the short budget runs (35 epochs).

Table 24: IWSLT'14 (DE-EN) on Transformer networks full budget training (50 epochs): Training, validation perplexity and test BLEU scores for more learning rate schedules. The test BLEU scores are computed on the checkpoint with the best validation perplexity. We report the mean and standard deviation over 3 runs.

| LR Schedule | Test BLEU Score | Train ppl | Validation ppl |
|---|---|---|---|
| One-Cycle | 34.77 (0.064) | 3.68 (0.009) | 4.97 (0.010) |
| Cosine Decay | 35.21 (0.063) | 3.08 (0.004) | 4.88 (0.014) |
| Linear Decay | 34.97 (0.035) | 3.36 (0.001) | 4.92 (0.035) |
| *Knee schedule* | **35.53** (0.06) | 3.00 (0.044) | 4.86 (0.02) |

### E.5 SQUAD-V1.1 FINETUNING WITH BERT$_{BASE}$

We choose 1e-5 as the maximum learning rate for One-Cycle runs as the minima occurs close to 1e-4 . For linear, cosine decays we start with a seed learning rate of 3e-5 as used in standard baselines. Table 26 show the average training loss, average test EM and F1 scores for the various schedules. We did not do a short budget training for this dataset, as the full budget is just 2 epochs.

Table 25: IWSLT'14 (DE-EN) on Transformer networks short budget training (35 epochs): Training, validation perplexity and test BLEU scores for more learning rate schedules. The test BLEU scores are computed on the checkpoint with the best validation perplexity. We report the mean and standard deviation over 3 runs.

| LR Schedule | Test BLEU Score | Train ppl | Validation ppl |
|---|---|---|---|
| One-Cycle | 34.43 (0.26) | 3.98 (0.028) | 5.09 (0.017) |
| Cosine Decay | 34.46 (0.33) | 3.86 (0.131) | 5.06 (0.106) |
| Linear Decay | 34.16 (0.28) | 4.11 (0.092) | 5.14 (0.066) |
| *Knee schedule* | **35.08** (0.12) | 3.58 (0.063) | 4.90 (0.049) |

Table 26: SQuAD-v1.1 fine-tuning on BERT$_{BASE}$ for more learning rate schedules. We report the average training loss, average test EM, F1 scores over 3 runs.

| LR Schedule | EM (av) | F1 (av) | Train Loss (av) |
|---|---|---|---|
| One Cycle | 79.9 (0.17) | 87.8 (0.091) | 1.062 (0.003) |
| Cosine Decay | 81.31 (0.07) | 88.61 (0.040) | 0.999 (0.003) |
| Linear decay | 80.89 (0.15) | 88.38 (0.042) | 1.0003 (0.004) |
| *Knee schedule* | **81.38** (0.02) | **88.66** (0.045) | 1.003 (0.002) |

# F DETAILED PLOTS

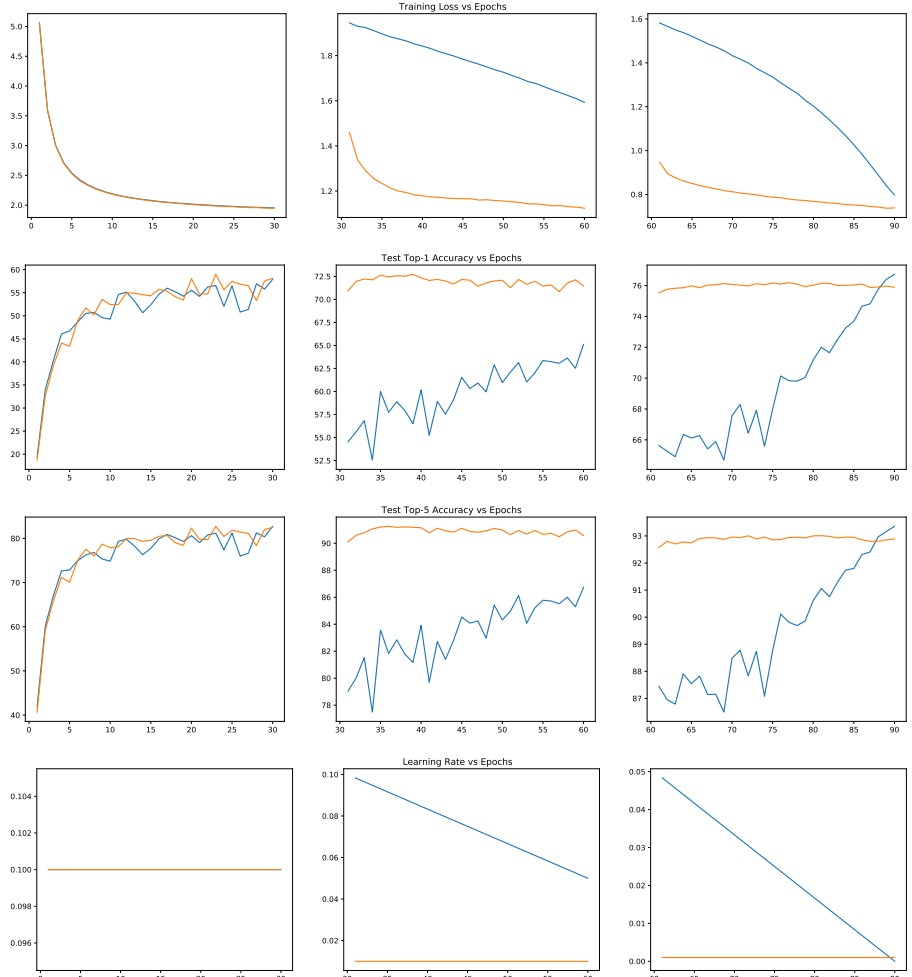

Figure 5: ImageNet on Resnet-50 trained with Momentum. Shown are the training loss, top-1/top-5 test accuracy and learning rate as a function of epochs, for the baseline scheme (orange) vs the *Knee schedule* scheme (blue). The plot is split into 3 parts to permit higher fidelity in the y-axis range.

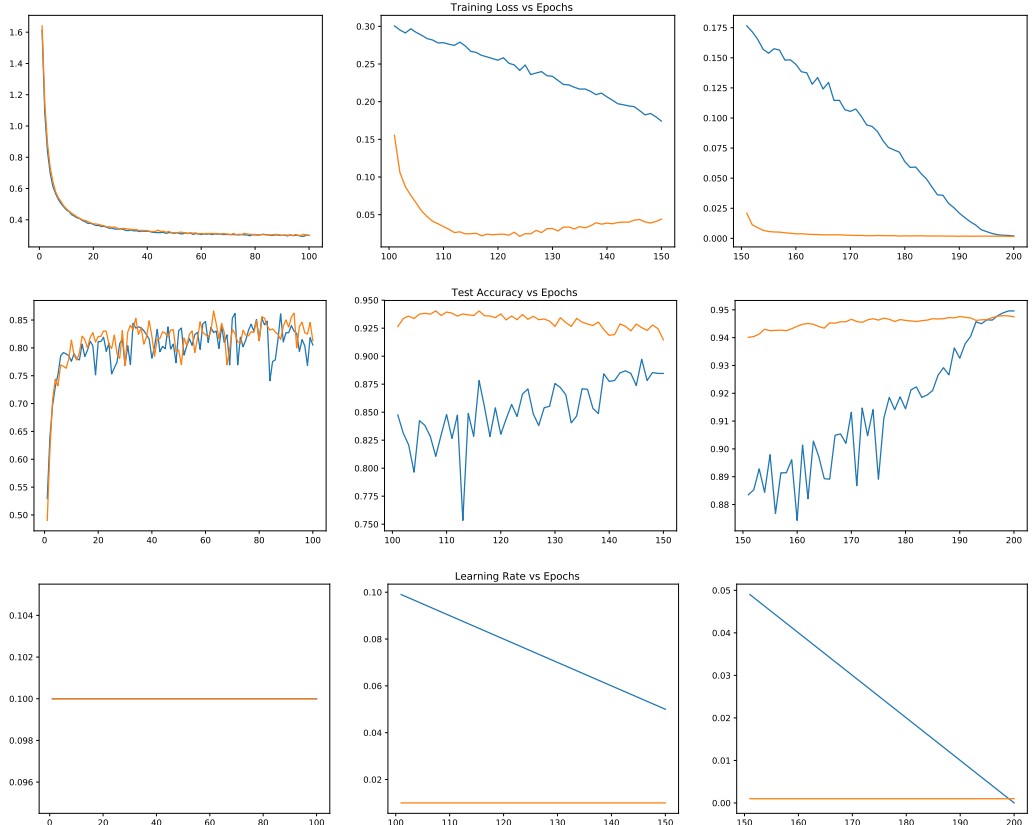

Figure 6: Cifar-10 on Resnet-18 trained with Momentum. Shown are the training loss, test accuracy and learning rate as a function of epochs, for the baseline scheme (orange) vs the *Knee schedule* scheme (blue). The plot is split into 3 parts to permit higher fidelity in the y-axis range.

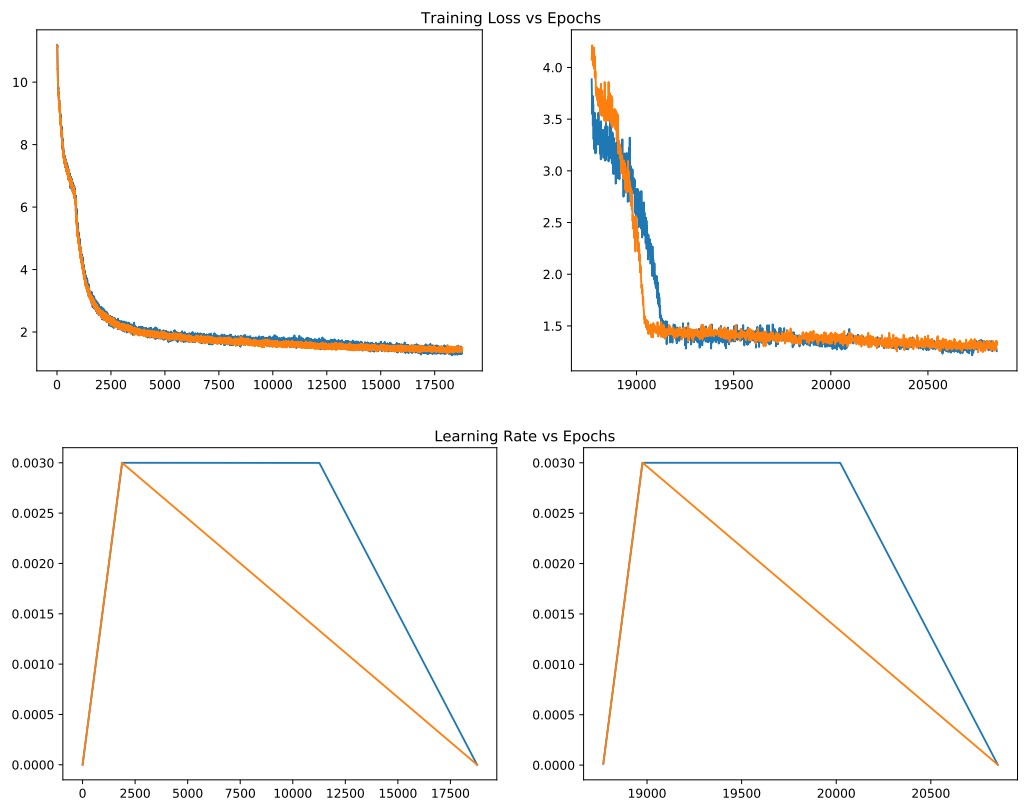

Figure 7: BERT$_{\text{LARGE}}$ pretraining for batch size of 16k with LAMB optimizer for the short budget runs. Shown are the training loss and learning rate as a function of steps, for the baseline scheme short budget (orange) vs the *Knee schedule* scheme short budget (blue). The plot is split into 2 parts to give a clear picture of the two phases of training Devlin et al. (2018). Note that even though the training loss curves look similar for the two runs, we see a significant gap in F1 score obtained when we fine-tune the model checkpoints on SQuAD-v1.1 Rajpurkar et al. (2016). See Table 27 for details.

| LR Schedule | F1 - Trial 1 | F1 - Trial 2 | F1 - Trial 3 | F1 avg. | F1 max |
|---|---|---|---|---|---|
| Baseline (short budget) | 90.39 | 90.64 | 90.53 | 90.52 | 90.64 |
| *Knee schedule* ( short budget ) | 91.22 | 91.29 | 91.18 | 91.23 | 91.29 |
| *Knee schedule* ( full budget ) | 91.45 | 91.41 | 91.51 | 91.46 | 91.51 |

Table 27: SQuAD fine-tuning on BERT$_{\text{LARGE}}$. We report F1 scores for 3 different trials as well as the maximum and average values.

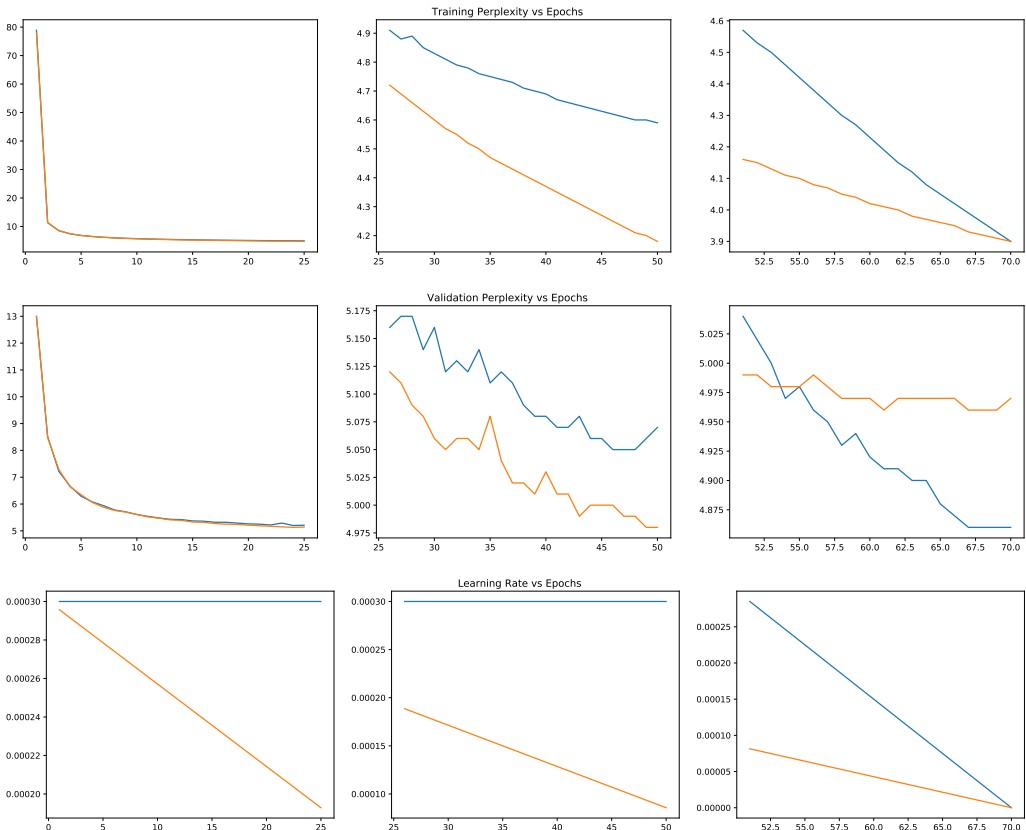

Figure 8: WMT'14 (EN-DE) on Transformer$_{BASE}$ network trained with RAdam. Shown are the training perplexity, validation perplexity and learning rate as a function of epochs, for the baseline scheme (orange) vs the *Knee schedule* scheme (blue). The plot is split into 3 parts to permit higher fidelity in the y-axis range.

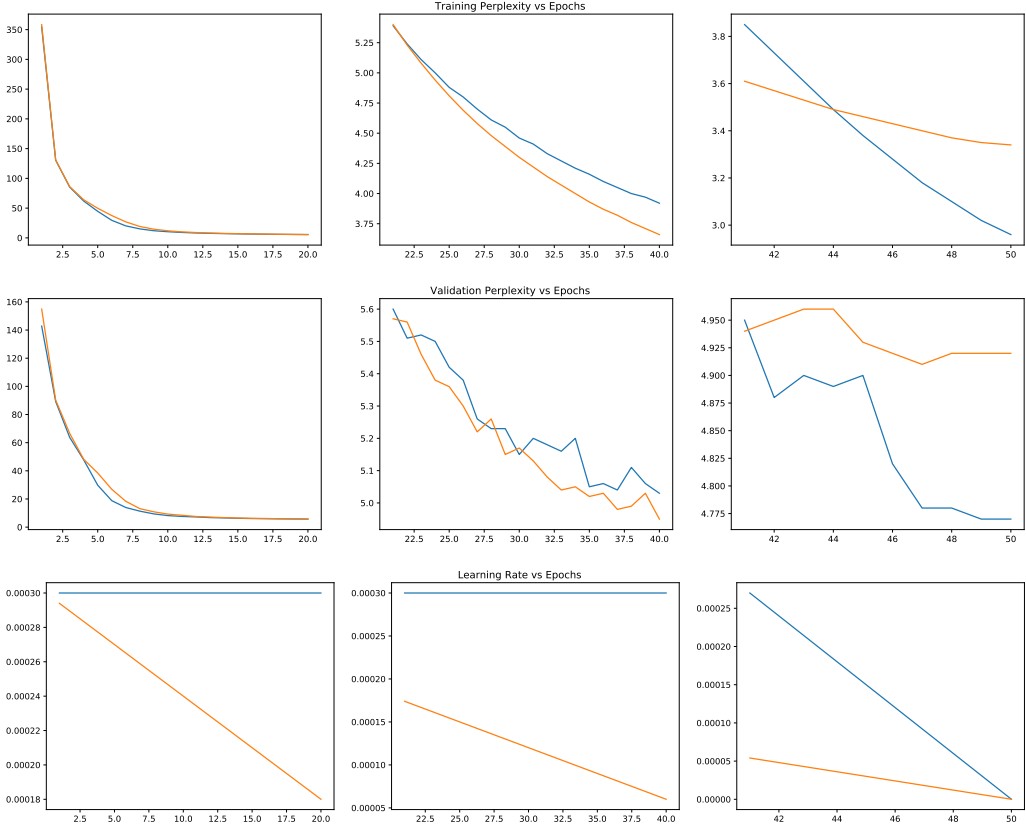

Figure 9: IWSLT'14 (DE-EN) on Transformer$_{BASE}$ network trained with RAdam. Shown are the training perplexity, validation perplexity and learning rate as a function of epochs, for the baseline scheme (orange) vs the *Knee schedule* scheme (blue). The plot is split into 3 parts to permit higher fidelity in the y-axis range.

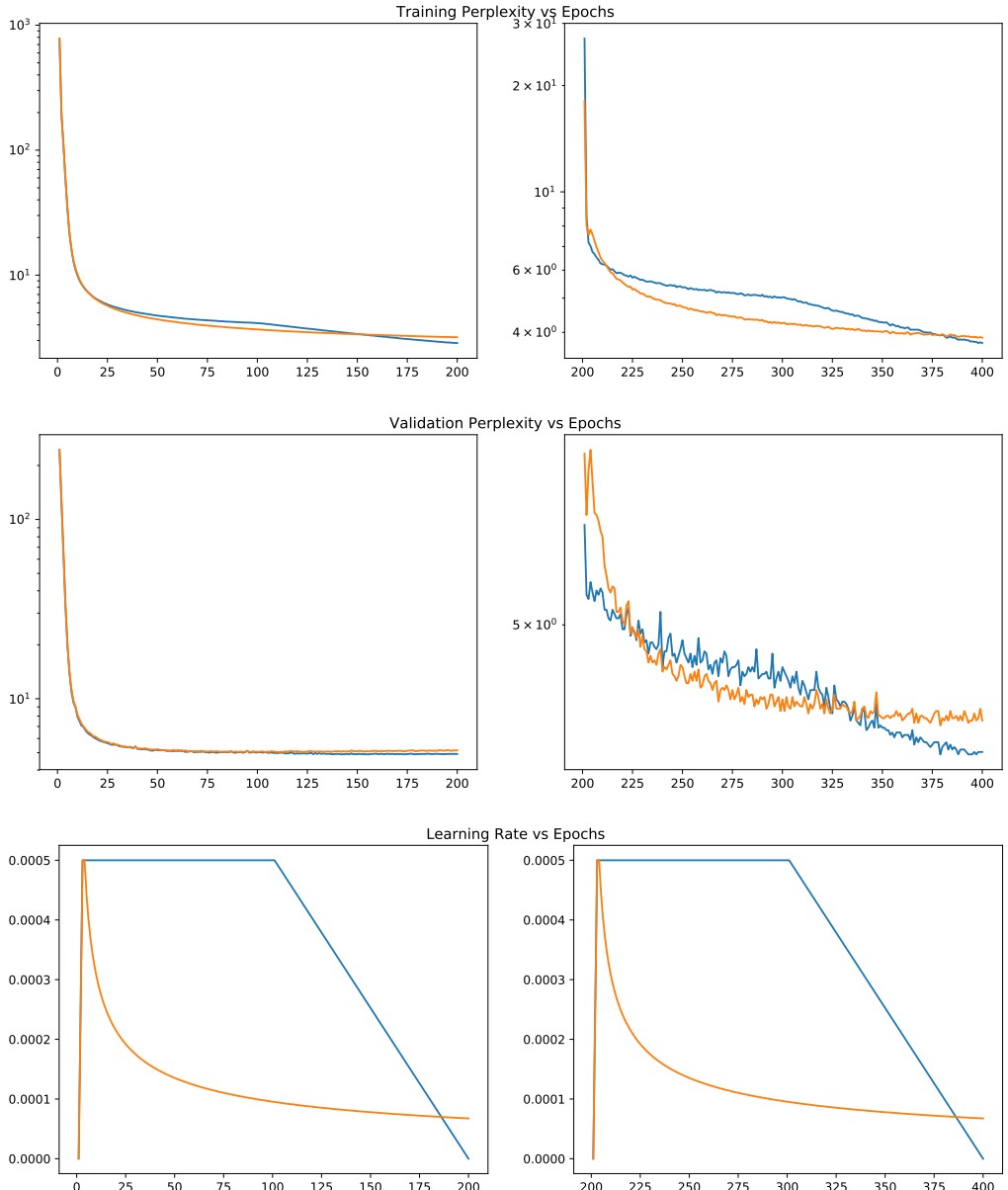

Figure 10: IWSLT'14 (DE-EN) on MAT network trained with Adam. Shown are the training perplexity, validation perplexity and learning rate as a function of epochs, for the baseline scheme (orange) vs the *Knee schedule* scheme (blue). MAT training involves two training phases with 200 epochs each, shown in separate columns above.

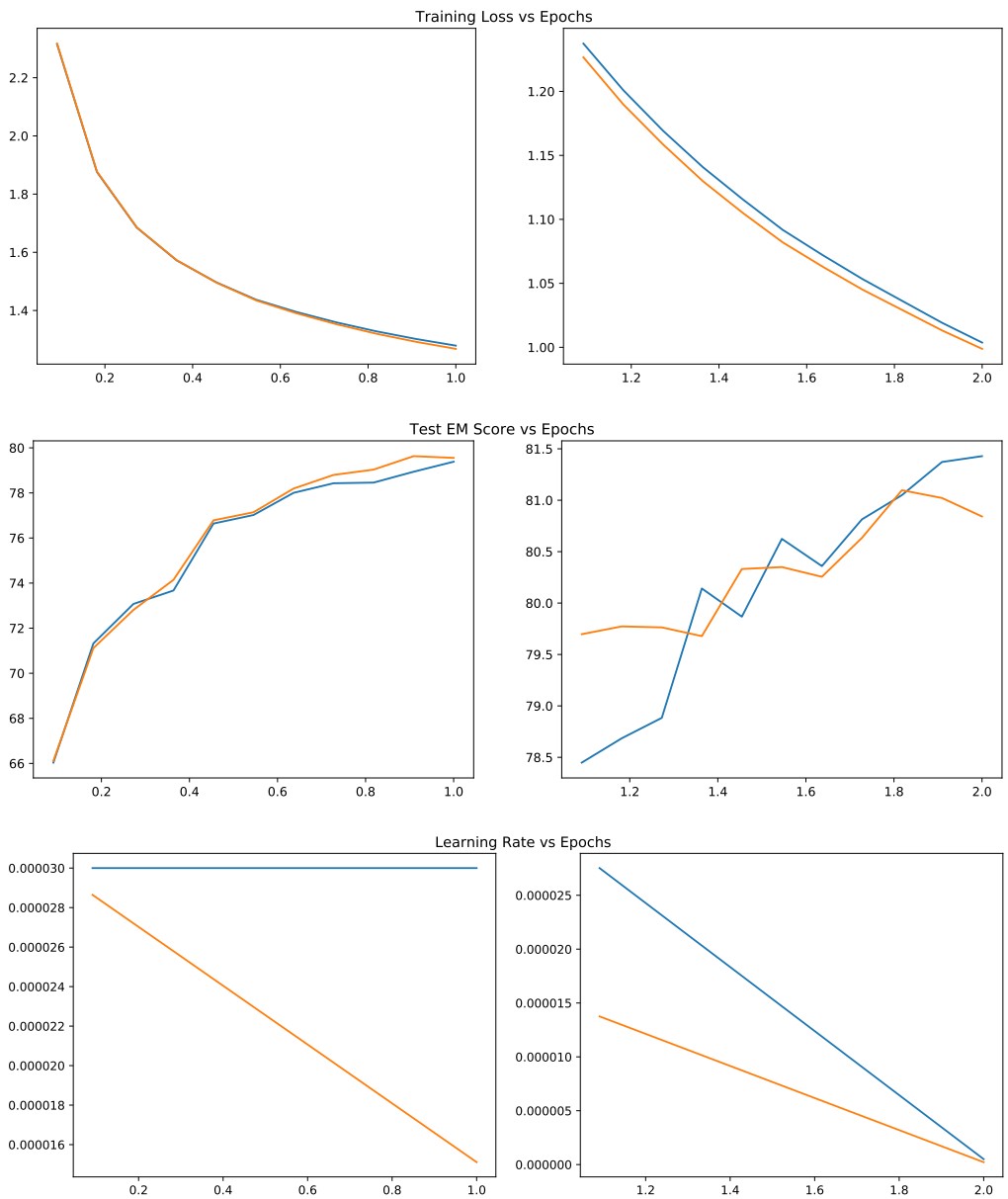

Figure 11: SQuAD-v1.1 fine-tuning on BERT$_{BASE}$ trained with Adam. Shown are the training loss, test EM score, and learning rate as a function of epochs, for the baseline scheme (orange) vs the *Knee schedule* scheme (blue). The plot is split into 2 parts to permit higher fidelity in the y-axis range. It is clear that with *Knee schedule* the network starts to overfit after the 2nd epoch, where the testing loss continues to go down, but generalization suffers. We saw similar behavior with different seeds, and thus need to train with *Knee schedule* for only 2 epochs.

