# OpenReview forum: "Wide-minima Density Hypothesis and the Explore-Exploit Learning Rate Schedule"
_ICLR.cc/2021/Conference — Reject_

### Official Review · AnonReviewer4 · 2020-10-28
**Unclear novelty and problematic baseline comparisons**

**Rating:** 3
**Confidence:** 5

**Review:**

Overview:
Overall I believe the comparisons to baselines seem too problematic to understand the value of the proposed method. Regarding the reduced training budget results (“Knee schedule can achieve the same accuracy as the baseline with a much reduced training budget”), were the baseline schedules also retuned for the reduced training budget? If not, this seems like an unfair advantage to the proposed method. For example, in the MLPerf competition (https://arxiv.org/abs/1910.01500) for ImageNet there have been schedules (consisting of a linear warmup followed by quadratic decay) that have been tuned to reach 75.9% in only 64 epochs, even at massive batch sizes, implying that the baseline schedule in Table 3 could likely do much better than what is reported if it was retuned with the same number of trials as the proposed method, or if a more competitive baseline schedule was used. Some of the results seem misleading as well; in the training curve figures 6, 7, 8, 9, 10 in the appendix, it seems odd that the proposed method only catches up to the (untuned) baselines towards the very end of training, and that this was not mentioned in the main text. For example:
-on CIFAR10 the baseline beats the proposed method until the final *5 out of 200* epochs of training
-on BERT_LARGE pretraining it is unclear from the plots when the proposed method beats the baseline as the curves are so similar
-on WMT’14 (EN-DE) the baseline beats the proposed method until the final 54 out of 70 epochs of training
-on IWSLT’14 (DE-EN) the baseline and proposed method cross each other a few times, the final time being at epoch 41 of 50
-on IWSLT’14 (DE-EN) with the MAT network, the baseline and proposed method cross each other a few times, the final time being at epoch 330 of 400
While it is not invalid for a proposed method to overtake a baseline towards the end of training, these results indicate that perhaps if the baselines were retuned, they could maintain their better performance for the last few epochs of training. Using the same initial LR for the proposed and baseline methods is useful, however it is insufficient to demonstrate that the proposed method could still perform well under different initial conditions. I have additional concerns about the significance of the proposed method over the baselines which I describe below.

Regarding comparing to the sharpness of the baseline LR schedules: “With fewer explore epochs, a large learning rate might still get lucky occasionally in finding a wide minima but invariably finds only a narrower minima due to their higher density.“ it would help to show curvature metrics at frequent intervals during training to confirm this hypothesis, and to also show these for the other learning rate schedules compared to, so that you can demonstrate that the proposed schedule achieves something the baselines cannot. The sharpness values in Figure 2 are interesting, but I am unable to determine how impressive they are given that they are not compared to sharpness values for any other schedules, so I don’t know what the baseline numbers should be.

Finally, it is unclear that the proposed method is novel enough to warrant a standalone paper, without more rigorous theoretical explanations to support the claimed reasons behind its performance.

Pros:
-It is useful to note that definitions of curvature can be problematic, which the authors do discuss (citing https://arxiv.org/abs/1703.04933)
-The breadth of experiments is genuinely impressive, but unfortunately would be more impressive if the breadth was smaller and more careful tuning was done for the proposed method and baselines

Concerns:
-In Appendix C when describing your curvature metric, you say “The maximization problem is solved by applying 1000 iterations of projected gradient ascent”. How was 1000 chosen? Did the sharpness metric stop changing if more steps were used?
-What are the stddevs of the results in Tables 6, 7, 18, 19, 20, 21, 26? The proposed results seem very close to the (untuned) baselines, and so it would be useful to understand how statistically significant they are.
-Toy problems can be extremely useful to empirically demonstrate this wide vs sharp minima selection phenomena would be useful, such as in Wu et al. 2018 (https://papers.nips.cc/paper/8049-how-sgd-selects-the-global-minima-in-over-parameterized-learning-a-dynamical-stability-perspective), or the noisy quadratic model in https://arxiv.org/abs/1907.04164
-The curves in Figure 7 seem extremely similar, it would help to plot the loss on a log scale
-In Section 4.1, “In all cases we use the model checkpoint with least loss on the validation set for computing BLEU scores on the test set.”, is early stopping used in all experiments? If not, why?
-”Thus, in the presence of several flatter minimas, GD with lower learning rates does not find them, leading to the conjecture that density of sharper minima is perhaps larger than density of wider minima.” it is unclear to me how their previous results support this hypothesis; couldn’t one retune the learning rate of SGD to find sharper/flatter minima, independently of how many sharp/flat minima exist?

Writing:
-The experiment details in the intro could be moved to later in the paper (it seems to be repeated in section 2)
-Overall the paper length seems like it could be drastically reduced by removing repeated statements
-Figures 6, 7, 8, 9 would be much clearer to read if it was a single plot per row, possibly on a log scale on the vertical axis when applicable
-For consistency, it would be useful to have “Baseline (short budget)” also be reported in Table 5

Prior work:
There are many previous works on explaining the benefits of large learning rates, the most relevant being https://arxiv.org/abs/1907.04595 which seems to make the same case as this paper, but is not cited. Additionally, https://arxiv.org/abs/2003.02218 has more theoretically explanations for this, using the Neural Tangent Kernel literature, and the authors could likely derive similar explanations. In fact, they use a similar schedule as the proposed method, but do not give it a name: “The network is trained with different initial learning rates, followed by a decay at a fixed physical time t · η to the same final learning rate. This schedule is introduced in order to ensure that all experiments have the same level of SGD noise toward the end of training.” Finally, there are other works that describe how low curvature directions of the loss landscape will be learned first, benefiting from a higher LR, followed by high curvature/high noise directions, which benefits from a smaller LR, described in https://arxiv.org/abs/1907.04164. I believe that a more formal explanation and analysis of the claims on solution curvature density should be provided.

Additional feedback, comments, suggestions for improvement and questions for the authors:
I believe that fairer experimental setup would be similar to the following:
-pick several competitive LR schedules for each problem (not just the “standard” ones)
-identify a similar number of hyperparamters for each*, such as number of warmup steps, decay values, decay curve shapes, etc.
-retune each schedule and the proposed method for the same number of trials, using similarly sized search spaces for each (ideally one would also retune the initial/final learning rates, momentum, and other hyperparameters for each, but this may be too expensive)
-select the best performing hyperparameter setting for each schedule, and rerun it over multiple seeds to check for stability

*it can be problematic to make comparisons across methods with different numbers of hyperparameters even with the same tuning budget, because it is impossible to construct the same volume hyperparameter spaces with different numbers of hyperparameters, see https://arxiv.org/abs/2007.01547 for a more thorough treatment

---

> ### Author Response · Authors · 2020-11-17
> **Response to Reviewer #4 (part 1/2)**
>
> 1. Comparison with baselines:
>
>     * “were the baseline schedules also retuned for the reduced training budget”: For BERT and WMT, IWSLT transformer runs, linear decay was the baseline schedule which has no hyperparameters and thus required no tuning. Please see Table-7 where we outperform the linear decay baseline for these experiments.
>
>     * “MLPerf competition for ImageNet there have been schedules ….. that have been tuned to reach 75.9% in only 64 epochs”: We assume the reviewer is referring to https://arxiv.org/pdf/1909.09756.pdf  Note that, in order to achieve 75.9 in 64 epochs, they, apart from tuning the learning rate schedule, i) use the adaptive learning rate scaling LARS optimize, ii) change LARS update to use ‘unscaled’ momentum and iii) tune the momentum parameter. Thus, the comparison of Knee with MLPerf results is not apples-to-apples in so many dimensions, including of course the batch size.  We are happy to run the MLPerf’s quadratic decay on standard small-batch ImageNet training with SGD, but based on our experience, we do not expect it to perform as well as Knee.
>
> 2. “Some of the results seem misleading as well; in the training curve figures 6, 7, 8, 9, 10 in the appendix, it seems odd that the proposed method only catches up to the (untuned) baselines towards the very end of training, and that this was not mentioned in the main text.”
>     * We would request the reviewer to clarify what is “misleading” about this. It is well-known that higher learning rates don’t perform well earlier on in the training (in fact we talk about exactly this in our paper – see Figure 1). Since we run at a higher learning rate than the baselines during the initial part, the above behavior in the initial stages is *expected*. Only when the learning rate gets low during the end of the decay, the optimizer is able to get to the bottom of the wide minima and the test accuracies see a bump (See Figure 5 for example).
>
>     * Also, please can the reviewer explain why they say “untuned” baselines? Figures 6-10 compare full budget runs which use heavily tuned baselines which in most cases were used by the authors of the papers which achieved SOTA at the time of publishing (e.g., see the BERT pre-training experiment, or the MAT experiment where the baseline is close to SOTA currently, and we achieved a new SOTA on the IWSLT’14 (DE-EN) and WMT’14 (DE-EN) datasets).
>
> 3. “Using the same initial LR for the proposed and baseline methods is useful, however it is insufficient to demonstrate that the proposed method could still perform well under different initial conditions"
> Please see Appendix D where we evaluate the learning rate sensitivity of our schedule. We observed that the seed learning rate can impact the final accuracy, but Knee schedule is not highly sensitive to it (see Table 17). In fact, we can achieve higher accuracy than reported by tuning the seed LR as well. However, we did not do that due to time constraints.
>
> 4. Sharpness:
>     * “it would help to show curvature metrics at frequent intervals during training to confirm this hypothesis”: Unfortunately, the sharpness / curvature metrics make sense only near the minimum and not far away from it -- e.g. it doesn’t tell much if farther from the minimum we have a high/low curvature. We also verified this by empirically evaluating curvature at intermediate points to see if they can give an estimate of the curvature at the minimum or close to it, but we found that there is little correlation between them.
>
>     * “The sharpness values in Figure 2 are interesting, but I am unable to determine how impressive they are given that they are not compared to sharpness values for any other schedules, so I don’t know what the baseline numbers should be.”: Just to be clear, we did not use our Knee schedule for Figure 2 and used a step schedule with different durations for the high (explore) LR portion. Since all LR schedules don’t permit an explore duration tuning (e.g. linear/cosine decay), we are not sure how we will do the suggested evaluation on these baselines.
>
> 5. “The maximization problem is solved by applying 1000 iterations of projected gradient ascent”. How was 1000 chosen?”: We simply use a very large number so that the optimization converged.
>
> 6. “What are the stddevs of the results in Tables 6, 7, 18, 19, 20, 21, 26? “: Tables 18, 19, 20, 21, 26 already report the stddev in parenthesis. For Table 6, 7 the stddevs are mentioned in the detailed tables – Tables 3-5, 10-13 and 18-24. We will incorporate these into Table 6 itself if needed.
>
> 7. “The curves in Figure 7 seem extremely similar, it would help to plot the loss on a log scale”: Sure, we will update that.

---

> > ### Comment · AnonReviewer4 · 2020-11-20
> > **Unclear novelty and problematic baseline comparisons, response**
> >
> > Thank you for the detailed reply, I appreciate your taking the time to consider all my points.
> >
> > 1a. Couldn't you have tuned the initial/final learning rates? I believe it is unfair to tune your schedule on a shorter training budget and then compare to a learning rate schedule that was selected for a longer training budget, but perhaps I am still misunderstanding your procedure for generating the results in Table 4.
> > 1b. Sorry, I believe my reply was misleading. I mainly was interested in if you had retuned the baselines schedule in Table 3 for a 50 epoch training budget, and was giving the MLPerf LR schedule as an example of a schedule that possibly could serve as a more competitive baseline for a reduced budget (just using the schedule, not all the fancier tricks, would likely cut down training time below 90 epochs).
> >
> > 2a. My concern is that because the proposed method only catches up to the baseline in the final bit of training, if a slightly better training schedule were used then the proposed method may never actually beat the baseline, whereas if it dominated the baseline for a longer duration then perhaps the comparison would be less sensitive to selection of baseline. For example if you retuned the baseline CIFAR schedule in Figure 6, you could very likely increase the final accuracy by 0.2%, which would make your proposed method indistinguishable in terms of the final performance. It is a fair point that perhaps the proposed method may not perform well until the LR is sufficiently small; did you consider decaying the LR to a small value earlier in the training budget, and then training with a smaller (possibly also decaying?) LR for the rest? This could more quickly beat the baselines. I understand your experiments where you cut the training budget are similar to this, but again the baseline LR schedules were not always retuned for these shorter budgets so we don't have an applies-to-apples comparison.
> > 2b. In general I believe that the baseline schedules should be retuned. While it is true that previous authors are also aiming for SOTA, they perhaps did not use as much compute to tune as the proposed method, or perhaps they did not care to tune further after achieving their goal. Either way retuning ensures a fair comparison is being made.
> >
> > 3. That is a useful plot for the proposed schedule, however we do not know whether or not this is also the case for the baseline schedules.
> >
> > 4a. I believe that in general it would be useful to show some notion of curvature away from the final minima, because this could tell us if we are indeed still exploring during the explore phase of the proposed schedule. Ideally we would see a change in curvature once we start decaying the LR.
> > 4b.  One could calculate the sharpness every 500 steps during training, for the proposed and baseline schedules, and this should show whether or not the proposed method is actually performing a uniquely useful exploration/exploitation behavior.
> >
> > 5. Thank you for clarifying!
> >
> > 6. I misread the stddev caption on these tables initially, thank you for correcting my mistake.
> >
> > 7-10. Thank you!
> >
> > 11. I believe that tuning the seed LR could have a dramatic affect on baseline performance.
> >
> > 12. Responded to the common comment.
> >
> > While the breadth of experiments is impressive in this paper, I am still not convinced that the results are compared to competitive baselines, in addition to my novelty concern. I unfortunately do not seem myself raising my score, but I encourage the authors to retune their baselines and provide more detailed analyses (perhaps even some theoretical ones on linear models?) and resubmit this to a future venue!

---

> ### Author Response · Authors · 2020-11-17
> **Response to Reviewer #4 (part 2/2)**
>
> 8. “is early stopping used in all experiments? If not, why?”: It is used in experiments where a separate test and validation set was available. Both baselines and knee used the same policy for all experiments. As reported in the paper “For all experiments, we used an out of the box policy, where we only change the learning rate schedule, without modifying anything else.”
>
> 9. Writing: Thank you for the feedback. We will incorporate the suggestions.
>
> 10. Prior work: Thank you for the suggestions. We will add a discussion on these.
>
> 11. Experimental setup: Our approach for evaluation has been to take papers which reported SOTA (or close to SOTA) results at the time of publishing and had reasonable compute requirements, and use the LR schedules used in those papers as our baselines. Since learning rate tuning (both tuning the schedule params and trying different schedules) is almost always done to achieve the best results, we regard our baselines to be competitive. That said, we do want to point out that in many experiments, such as BERT pre-training, the WMT, IWSLT translation tasks the competitive baselines were linear decays which do not have a tuning parameter other than the seed LR.
>
> 12. Limited novelty: Please see common comment titled "Response to reviewers (common)"

---

### Official Review · AnonReviewer1 · 2020-10-29

**Rating:** 4
**Confidence:** 2

**Review:**

Summary:
This paper did an empirical study on the learning rate (LR) schedule for deep neural networks (DNNs) training. The authors argue that the density of wide minima is lower than sharp minima and then show that this makes keeping high LR necessary. Finally, they propose a new LR schedule that maintains high LR enough long.

Pros:
-	The problem this paper studies is import for DNNs training. The proposed LR schedule is simple and has the potential to be used widely.
-	The authors conduct extensive empirical tests to support their claim and the experimental design is reasonable.

Cons:
-	I’m not fully convinced by the hypothesis that wide minima have lower density. The empirical results can be explained by other hypotheses as well. For example, it is also possible that wide minima are farther away from the initialization. I think the authors need to either provide theoretical analysis or come up with new experiments to further verify this hypothesis.
-	The proposed LR schedule does not seem necessary. One could easily achieve the same purpose by existing LR schedules, e.g. use a step decay LR schedule.
-	The novelty is low. The main novelty of the paper is the above hypothesis, but it is not supported enough. The proposed LR schedule is a slightly modified version of the existing LR schedule.  Thus the contribution of this paper seems incremental.

---

> ### Author Response · Authors · 2020-11-17
> **Response to Reviewer #1**
>
> 1. Hypothesis validation: “For example, it is also possible that wide minima are farther away from the initialization.” -- we already performed an experiment to rule this out (please see Section 2 – second last paragraph on page-3), where we ran very long training experiment (cifar10 with 10000 epochs or 50 times the normal training run of 200 epochs) with a low LR to allow it enough time to reach farther from initialization. Although the training converged, the final test accuracy was much worse than knee schedule on the 200 epoch run.
> Apart from this, we did several other empirical studies to validate the hypothesis, and found that the hypothesis could explain all the observations. For example, see Figures 2, 3 where we predict the qualitative distribution of accuracies and sharpness from our hypothesis and observe the same experimentally. We also detail more experiments in Appendix A on a different dataset and learning rate schedule and had similar observations.
> We believe that we have done rigorous empirical verification of our hypotheses, but would be happy to do more experiments and welcome suggestions from the reviewers.
>
> 2. “The proposed LR schedule does not seem necessary. One could easily achieve the same purpose by existing LR schedules, e.g. use a step decay LR schedule.”: This is actually invalidated in our experiments, where we see that knee schedule gives a bump over such baselines (e.g. see Table 3 where Knee has a 0.8% absolute gain on top-1 accuracy on ImageNet dataset over the step schedule baseline). Moreover, our hypothesis enables a more principled design of learning rate schedules rather than the heuristic approaches used today.
>
> 3. Limited novelty: Please see common comment titled "Response to reviewers (common)"

---

> > ### Comment · AnonReviewer1 · 2020-11-24
> > **Thank you for your response**
> >
> > - "we ran very long training experiment with a low LR to allow it enough time to reach farther from initialization" Did you verify it is far from the initialization? With a low LR, the model could be trapped in a local minimum so running longer may not affect the distance from initialization.
> >
> > - In Table 2, if you continue increasing the number of exploring epochs for step decay, will the performance improve? Have you tried a larger LR for the second stage of step decay? Because by looking at Figure 5, it seems like the improvement of the proposed LR schedule over step decay comes from using a larger LR after the first decay. I wonder if tuning the second stage of step decay can achieve a similar effect as the proposed LR schedule.

---

> > > ### Author Response · Authors · 2020-11-25
> > > **Response to questions**
> > >
> > > 1. We will compute the delta-w and add it to the paper but our past experience with computing delta-w across runs suggest that i) there is little correlation between delta-w and accuracy and ii) running longer generally increases delta-w, most likely due to the high dimensional landscape.
> > >
> > >
> > > 2. In table-2 the improvement from increasing the number of fine-scale explore stagnates eventually. The issue with changing LR for second stage of the step schedule is that the hyper-parameter search space is very large since we have to identify both LR and step duration. One simplification may be to keep step duration the same and vary LR -- we are happy to run these experiments and add them to the paper. From experience, we found that linear decay worked well mainly because it was able to do the finerscale “exploit” at all learning rates rather than a fixed LR of step schedule.

---

### Official Review · AnonReviewer3 · 2020-10-31
**Interesting Observations But Limited Novelty**

**Rating:** 5
**Confidence:** 3

**Review:**

Learning rate schedule plays an important role in DL, which has a large influence over the final performance. Though there have been lots of schedules, achieving SOTA performance still requires careful hand-tuned schedule that may be case by case. Compared with previous learning rate schedules, authors first conjectured that the number of wide minima is significantly lower than the number of sharp minima, and then  proposed to use a large learning rate at the initialization phase for sufficient exploration to achieve a wide minima, which may achieve better generalization performance. Extensive experiments validate the proposed learning rate schedule.

The observation of this paper looks interesting, and authors have conducted lots of experiments to validate the effects of proposed learning rate schedules. However, the novelty of this paper seems limited. First, authors conjecture that the number of wide minima is significantly lower than the number of sharp minima, but it lacks a thorough investigation of this conjecture, either from related empirical study or theoretical understanding. Second, for the proposed learning rate schedule, it seems not very clear how to set the duration of exploration epochs appropriately across different tasks, as it is still a hand-tuned hyper-parameter. For fixed 50% explore, there is not much difference in terms of the performance compared with previous schedule such as Cosine Decay or linear decay in Table 6.

Overall, I tend to a weak reject and it would much better if authors could go deeper behind the observation/conjecture.

---

> ### Author Response · Authors · 2020-11-17
> **Response to Reviewer #3**
>
> 1. Validation of our hypothesis / conjecture: We did multiple empirical studies to validate the hypothesis, and found that the hypothesis could explain all the observations. For example, see Figures 2, 3 where we predict the qualitative distribution of accuracies and sharpness from our hypothesis and observe the same experimentally; we ruled out the argument that larger distance covered by large learning rates is the reason for better performance by running a very long training with a low LR which gave much worse performance (see Section-2 where we run Cifar10 for 10000 epochs, 50 times the typical training duration). We did more experiments in Appendix A on a different dataset and learning rate schedule and had similar observations.
> We believe that we have done rigorous empirical verification of our hypotheses, but would be happy to do more experiments and welcome suggestions from the reviewers. We acknowledge that a theoretical analysis of this phenomenon would help move our ‘hypothesis’ towards a ‘theory’ of wide minima in deep learning but that is outside the scope of this paper.
>
> 2. Explore hyperparameter: Yes, this is a hand tuned parameter, however we found this to be quite easy to tune as typically higher explore helps to a point after which it starts hurting. Thus, a simple binary search suffices.
> We are also exploring principled methods for automatically finding the optimal explore duration, by estimating the “width” of the landscape during explore phase and switching to exploit as soon as we identify the landscape is wide enough. Getting this estimate, however, is tricky given high dimensionality of DNNs and since intermediate points in the landscape may be far from the minimum.
>
> 3. Limited novelty: Please see common comment titled "Response to reviewers (common)"

---

### Official Review · AnonReviewer2 · 2020-11-02
**New learning rate schedule for training deep models leading to better generalization.**

**Rating:** 6
**Confidence:** 3

**Review:**

This work studies the problem of how to define learning rate schedules when training deep models so that the models better generalize.  To this end, the paper proposes and evaluates a learning rate schedule that consists of two stages (knee schedule).  A first stage of exploration adoptes a high learning rate. This initial stage is followed by a second stage where the learning rate decreases in a linear way. Extensive experimental results, both in text and image data, show that the proposed scheme allows one to train faster or to obtain better results with a fixed computational budget. The proposed learning schedule leads to SOTA results on IWSLT’14 (DE-EN) and WMT’14 (DE-EN) datasets.

The work relates the good performance of the proposed knee schedule in the hypothesis that wide minima have a lower density (are less common), therefore, a large learning rate  is required initially (and for some time) to avoid shallow minima. The second refinement stage with the learning rate declining linearly allows one to delve into the minimum found in the exploration stage. Recent works indicate that in fact wide minima are the ones that lead the models to generalize better and this is in agreement with the experimental results of the article.

The main contribution of the article is an exhaustive experimental evaluation in different applications where they analyze different schedules and show how the proposed schedule leads to superior performance. The paper raises a working hypothesis compatible with the success of the LR schedule and in that sense generates an interesting line to continue research.

Some questions:

1. From reading the article it is not clear to me how it is justified to keep the learning rate high even when the loss stagnates. I understand this is based on conducting experiments and then measuring the power of generalization. But it is interesting that from the training point of view it would seem that after training stagnates the network is not learning but pivoting from one side to the other. What do you think can be a good hypothesis of what is happening during training at this stage? I would like if possible that this point is better discussed. And it would also be useful if the work better discussed why the working hypothesis is the most reasonable explanation.

2. Table 1 shows that reducing the learning rate after the exploration stage helps to better minimize the loss. However, this does not translate into a network that generalizes better. Is it reasonable to hypothesize that during this second period the network overfitted to the behavior around this minimum? Does this phenomenon occur in other experiments? If so, why is the second refinement stage needed?

3. Warmup. Some optimizers use a warm up step where the learning rate starts to rise smoothly. It would be interesting to better discuss how this stage is linked to the exploration stage. How long does the warmup stage need to be? If warmup + exploration + decay is put together, at the end it is a curve with a certain resemblance to a cosine.

Additionally, if the information is available it would be useful to have the standard deviations of the average values ​​calculated in Table 6.

---

> ### Author Response · Authors · 2020-11-17
> **Response to Reviewer #2**
>
> 1. Justification for keeping LR high even when loss stagnates: According to our hypothesis the density of narrow minima is much higher than that of wide ones. Thus, with a high probability, the optimizer will find itself in a narrow minimum in the initial phase of training. Training at a high enough LR will allow the optimizer to “jump” out of these narrow minima as the step size would be large enough to come out of them. However, once it reaches a wide enough minimum, it will get stuck there. We thus train the optimizer long enough, so that it gets “stuck” in such a wide minimum. We do discuss this aspect in section 2, but will add more details.
> Is our hypothesis the most reasonable explanation: This is hard to say definitively, but the simplicity of our hypothesis and the fact that it explains all the observations of our experiments gives us confidence. We also invoke the Occam's razor in this regard.
>
> 2. Table 1: In our experiments, higher explore (up to a point) helped with improving the generalization performance. We saw this behavior consistently across all experiments. The first explore stage is needed to land in a wide minimum region, and the second exploit stage is needed to descend to the bottom of this wide minimum region.
>
> 3. Warmup relationship: Warmup is mostly needed for stability reasons, either because optimizers such as Adam have high initial variance (see [1]), or because the learning rate is scaled too high to accommodate high batch sizes. Without warmup, the training procedure typically becomes unstable. We treat warmup as orthogonal to our method, and simply use the same warmup duration as used by baselines. You are right that warmup + exploration + decay has resemblance to cos [-pi/2, pi/2]. Similarly, exploration + decay has resemblance to cos [0, pi/2].
>
> 4. Table 6 standard deviations: These are mentioned in the detailed tables – Tables 3-5, 10-13 and 18-24. We will incorporate these into Table 6 itself if needed.
>
> [1] Liyuan Liu, Haoming Jiang, Pengcheng He, Weizhu Chen, Xiaodong Liu, Jianfeng Gao, and Jiawei Han. On the variance of the adaptive learning rate and beyond. arXiv preprint arXiv:1908.03265, 2019

---

### Author Response · Authors · 2020-11-17
**Response to reviewers (common)**

We thank the reviewers for their comments.

Limited novelty: One common comment from several reviewers is that the solution proposed in the paper is too simple or that there is a lack of novelty. We would like to point out that while the knee schedule may be simple, it provides enormous value to researchers and practitioners in terms of higher test accuracy and/or reduced training time. For example, we show that substituting a complex inverse-square-root decay by the simpler knee schedule can help achieve a *state-of-the-art* result, a non-trivial outcome that few papers achieve. Similarly, while ResNet/Imagenet is typically trained with a complicated step schedule over 90 epochs (that requires tuning multiple hyper-parameters such as number of steps, length of each step, decay size between steps, etc.), a simpler knee schedule can achieve the same 90 epoch accuracy in 50 epochs, or almost half the time. Thus, we believe that the simplicity of knee is an advantage, not a drawback, and the community can benefit immensely from using a simpler knee schedule instead of complex schedules like inverse-square-root or step decay that don’t perform as well.

---

> ### Comment · AnonReviewer4 · 2020-11-20
> **Lack of novelty**
>
> While I agree that simpler is better, this is not an argument against lack of novelty. As far as I can tell, ML researchers and practitioners have been using schedules similar to this for a while, but have not tried to label it with a name. See https://paperswithcode.com/method/linear-warmup-with-linear-decay for a collection of papers that have done this (additionally with an LR warmup which usually seems to improve performance, which you did mention in the draft).
>
> While I am willing to forgo novelty to provide analyses of why commonly used techniques work, I don't believe this draft shows any additional insights into this schedule, besides the unverified hypothesis in section 2. If the sharpness analysis is done for baseline LR schedules (and perhaps other decay curves), and this verifies that a constant LR followed by a linear decay results is a superior way to explore then exploit local minima, this could be a useful explanation for the behavior.
>
> Additionally, I don't see how an inverse-square-root decay is any more complex than this schedule, assuming you use a square root and don't tune any of the exponents. One could view the proposed schedule as a special case of a polynomial decay with an exponent of 1.

---

> > ### Author Response · Authors · 2020-11-24
> > **Knee schedule is neither well-known nor commonly used**
> >
> >
> > We don't agree that we are simply labelling a well-known schedule as knee in this paper. The link you provide is simply a link to papers that use the well-known linear decay schedule (warmup is an *independent* aspect that is not the focus of our paper).  Similarly, your comment "One could view the proposed schedule as a special case of a polynomial decay with an exponent of 1." is again suggesting that knee is simply linear decay, which is incorrect.   Yes, linear decay is used in  the exploit phase of knee but, as we show in the paper, the explore phase of knee that uses a high and constant learning rate (lr) makes a huge difference in terms of final test accuracy.  Can you please point to actual papers that use a schedule 'similar' to knee that includes *both the explore and exploit phases*?
> >
> > Another piece of evidence that substantiates the claim that a knee-like schedule is not common/popular is to look at lr schedules that are supported by well known frameworks. Pytorch, for example, has a number of in-built lr schedulers in its torch.optim.lr_scheduler class (https://pytorch.org/docs/stable/optim.html) such as StepLR, MultiStepLR, ExponentialLR, CosineAnnealingLR, ReduceLROnPlateau, CyclicLR (which includes the triangularLR shape of linear increase/linear decrease that you point in the link above), OneCycleLR, etc. but nothing that is 'similar' to knee.    Similarly, huggingface, a popular repository for training language models has a number of lr schedules supported by default (https://huggingface.co/transformers/main_classes/optimizer_schedules.html) but nothing that is 'similar' to knee. Sure, one can code up the knee schedule without using a separate scheduler class but such an argument is applicable to most of the above schedules as well. Another indicator that knee is not popular/well-known is that not a *single baseline schedule* for any of the popular deep learning tasks that we have evaluated in the paper used a schedule that is 'similar' to knee.
> >
> > Thus, we believe that knee schedule is neither well-known nor commonly used. Knee may be 'similar' to well-known schedules from an abstract point of view (just like, say, RELU is 'similar' to sigmoid) but the actual learning rate values used in each step in knee is different enough that knee is able to significantly outperform all other well-known schedules in terms of higher final test accuracy, including achieving SOTA results in two tasks!

---

### Decision · Program_Chairs · 2021-01-07
**Final Decision**

**Decision:**

Reject

**Comment:**

The reviewers are concerned about the novelty of the proposed learning rate schedule, the rigor of the empirical validation, and the relationship between the results and the discussion of sharp vs. local minima. I invite the authors to incorporate reviewers' comments and resubmit to other ML venues.